# Coherent X-rays reveal anomalous molecular diffusion and cage effects in crowded protein solutions

Anita Girelli [1,2] ✉, Maddalena Bin [1], Mariia Filianina [1], Michelle Dargasz [3], Nimmi Das Anthuparambil [3], Johannes Möller [4], Alexey Zozulya [4], Iason Andronis [1], Sonja Timmermann [3], Sharon Berkowicz[1], Sebastian Retzbach [2], Mario Reiser[1], Agha Mohammad Raza [3], Marvin Kowalski [3], Mohammad Sayed Akhundzadeh[3], Jenny Schrage[3], Chang Hee Woo[5], Maximilian D. Senft [2], Lara Franziska Reichart[2], Aliaksandr Leonau[3,4], Prabhu Rajaiah Prince [6,7], William Chèvremont[8], Tilo Seydel [9], Jörg Hallmann [4], Angel Rodriguez-Fernandez [4], Jan-Etienne Pudell [4], Felix Brausse[4], Ulrike Boesenberg[4], James Wrigley [4], Mohamed Youssef [4], Wei Lu[4], Wonhyuk Jo[4], Roman Shayduk[4], Trey Guest[4], Anders Madsen [4], Felix Lehmkühler [6,10], Michael Paulus[5], Fajun Zhang [2], Frank Schreiber [2], Christian Gutt [3] & Fivos Perakis [1] ✉

Understanding protein motion within the cell is crucial for predicting reaction rates and macromolecular transport in the cytoplasm. A key question is how crowded environments affect protein dynamics through hydrodynamic and direct interactions at molecular length scales. Using megahertz X-ray Photon Correlation Spectroscopy (MHz-XPCS) at the European X-ray Free Electron Laser (EuXFEL), we investigate ferritin diffusion at microsecond time scales. Our results reveal anomalous diffusion, indicated by the non-exponential decay of the intensity autocorrelation function $g_2(q, t)$ at high concentrations. This behavior is consistent with the presence of cage-trapping between the short- and long-time protein diffusion regimes. Modeling with the $\delta\gamma$-theory of hydrodynamically interacting colloidal spheres successfully reproduces the experimental data by including a scaling factor linked to the protein direct interactions. These findings offer insights into the complex molecular motion in crowded protein solutions, with potential applications for optimizing ferritin-based drug delivery, where protein diffusion is the rate-limiting step.

In living organisms, proteins are embedded in crowded environments such as in the cytoplasm of cells, membranes, and lipid vesicles[1]. Understanding protein diffusion in these environments is essential, as it directly impacts several critical cellular processes, including metabolism (connected to reaction rates), self-assembly of supramolecular structures and signal transduction[1,2].

However, predicting and studying protein diffusion in crowded conditions is challenging due to the complex interplay of multiple factors. One such factor is the hydrodynamic interactions, where the motion of each biomolecule is influenced by the flow field generated by its neighbors[3,4]. These hydrodynamic flows can affect the entire biomembrane and enhance the diffusive motion of passive particles

within the cytoplasm[5]. Additionally, protein diffusion is affected by direct forces, including electrostatic interactions and nonspecific attractive interactions between molecules such as van der Waals forces[6]. Excluded volume effects can further complicate diffusion by reducing macromolecular mobility, although these effects alone do not fully account for the tenfold decrease in diffusion rates observed in vivo[7,8]. Moreover, transient protein clusters due to attractive forces can further diminish mobility[6,9]. A major challenge in understanding protein diffusion in crowded environments lies in deciphering the competition among these factors, all of which influence dynamics on similar time scales.

Crowding can also lead to deviations from simple Brownian motion, which is termed as anomalous diffusion[10,11]. For globular proteins and colloids, anomalous diffusion has been linked to cage effects[12,13], where a protein is transiently confined in a "cage" formed by neighboring molecules. The rearrangement of this cage is what eventually allows the molecule to diffuse and leads to structural relaxation. In this context, protein motion is typically characterized in terms of short- and long-time diffusion (see Fig. 1a). Short-time diffusion refers to the motion of proteins within the cage formed by its neighbors and occurs on time scales shorter than the interaction time $\tau_i$, which is the

travel time required for a protein to move a distance equal to its radius. The interaction time can be estimated as $\tau_i \approx \frac{R_h^2}{6D_s}$, where $R_h$ is the hydrodynamic radius of the protein and $D_s$ is the average self-diffusion coefficient. These two diffusion types are visible also by probing the mean square displacement (MSD) of the particles. They are characterized by a linear increase of MSD with time with one slope at $t \ll \tau_i$ and a different one at $t \gg \tau_i$[14].

Anomalous diffusion of globular proteins is most evident around the interaction time[3], $t \sim \tau_i$. At time scales much shorter than $\tau_i$ ($t \ll \tau_i$), the short-time diffusion is Brownian and is mainly affected by hydrodynamic interactions[15]. At much longer times ($t \gg \tau_i$), long-time diffusion is predominantly hindered by friction caused by interaction between the proteins[15]. Unlike short-time diffusion, there are no theoretical predictions for long-time diffusion due to the complex interplay of hydrodynamic and direct protein interactions. Therefore, it is essential to develop strategies to characterize the properties of the long-time diffusion, as well as the transition between the short- and long-time diffusion regimes.

To understand the role of hydrodynamic and direct interactions in long-time protein motion, it is necessary to measure the collective diffusion at length scales both larger and smaller than the protein

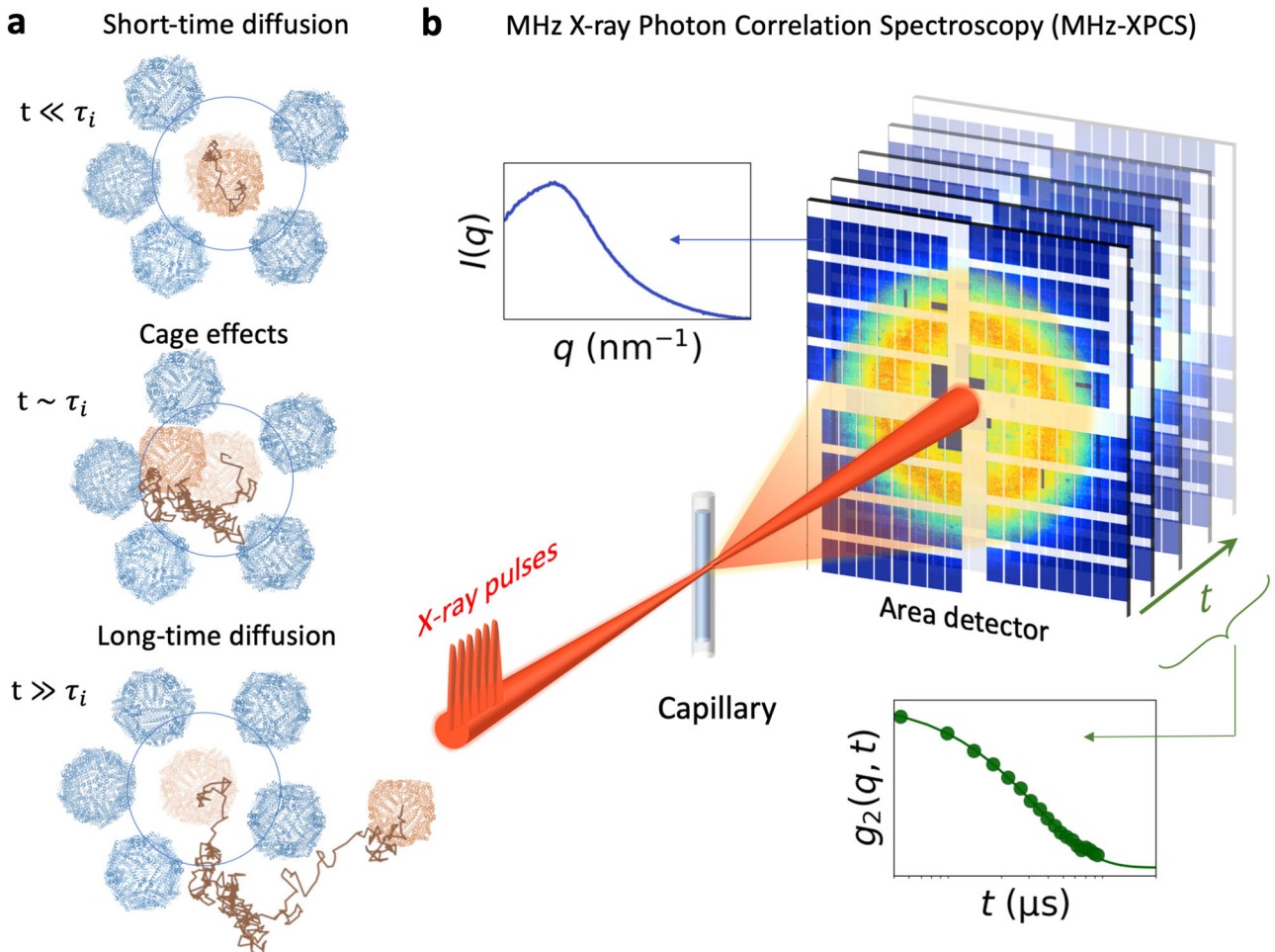

**Fig. 1 | Capturing protein diffusion across different length scales and time scales with MHz-XPCS. a** Conceptual representation of ferritin protein molecular dynamics at different time scales. The panels depict the short-time diffusion (top), cage effects (middle) and long-time diffusion (bottom). These different regimes are characterized by the interaction time, $\tau_i \approx R_h^2/(6D_s)$, where $R_h$ is the hydrodynamic radius and $D_s$ the self-diffusion coefficient. The ferritin images are rendered with Protein Data Bank (PDB file 6PXM [doi:10.2210/pdb6PXM/pdb]). **b** Schematic of the Megahertz X-ray Photon Correlation Spectroscopy (MHz-XPCS) experiment. Incident X-ray pulses are scattered from the ferritin protein solutions contained in a capillary. The scattered photons are detected per-pulse with an area detector (AGIPD 1M), capable of recording frames at MHz rate. The scattering intensity as a function of momentum transfer, $I(q)$, is obtained by azimuthal integration, while the intensity autocorrelation function $g_2(q, t)$ is determined by correlating the intensity across frames over time.

intermolecular distances, which is a significant technical challenge. Single particle tracking, fluorescence correlation spectroscopy[16], neutron backscattering, nuclear magnetic resonance (NMR) and gradient field NMR are limited to accessing the self-diffusion of proteins. While scattering techniques can probe collective diffusion, dynamic light scattering (DLS)[17,18] provides insights at length scales significantly larger than the protein size, while neutron spin echo reflects the short-time diffusion dynamics, typically faster than $\approx 300\,\mathrm{ns}$[19–22]. Thus, in addition to the theoretical challenges of describing long-time diffusion due to the complexity of inter-particle hydrodynamics and direct protein interaction, there is also an experimental research gap. Addressing this gap requires methods that can combine molecular-scale sensitivity with the capability to probe collective dynamics on microsecond time scales.

With the advent of high repetition rate X-ray free electron lasers (XFELs), such as the European XFEL (EuXFEL), it becomes possible to bridge this methodological gap. This capability is especially relevant for biological systems, enabling the study of relatively small proteins (on the order of several nanometers) at high volume fraction exceeding 20 vol%, which has been previously inaccessible. Measuring dynamics in the microsecond range opens the possibility to probe proteins around the interaction time $\tau_i$ (estimated in the order of few microseconds) and beyond. X-ray photon correlation spectroscopy (XPCS) enables direct access to collective protein diffusion on molecular length scales[23,24]. XPCS measurements have been demonstrated for measuring protein dynamics, both indirectly with microrheology[25], and directly when related to Brownian diffusion[26], cage relaxation[27,28], liquid–liquid phase separation dynamics[29–31], gelation processes[32–34], and nanoscale fluctuations in hydration water[35]. The recent development of Megahertz XPCS (MHz-XPCS) at EuXFEL has further enhanced the ability to capture equilibrium protein diffusion before the onset of radiation effects using the "correlation before aggregation" approach[36]. This technique offers a powerful tool to address the experimental research gap, enabling detailed investigation of long-time protein diffusion at the molecular scale and microsecond time scales, which we exploit in the present study.

Here, we investigate the short- and long-time molecular diffusion and cage effects in crowded protein solutions using MHz-XPCS at the EuXFEL (see Fig. 1b). The central questions are (i) how the interplay between hydrodynamic interactions and direct interactions influence the long-time protein diffusion, (ii) how long-time protein diffusion depends on the momentum-transfer $q$ approaching protein molecular length scales, (iii) what the characteristic signatures associated with the cage effects are in terms of time and length scales.

To address these questions, we utilize ferritin, a globular protein known for its non-toxic iron-storing capabilities, which is produced by nearly all living organisms. Ferritin has relevant applications in vaccine development[37,38], drug delivery[39] and nanomaterials[40,41]. Structurally, ferritin consists of a protein shell with 24 sub-units arranged in pairs to form a hollow nanocage, with a core enriched in iron ions. This robust and monodisperse system is ideal for dynamic measurements, providing a strong scattering signal as a result of the iron-rich core, which facilitates data interpretation. The expected interaction time $\tau_i$ for ferritin in water is $\tau_i \approx 0.3\,\mu s$ and $\tau_i \approx 3\,\mu s$ for ferritin in water-glycerol (with volume fraction $v_{glyc} = 0.55$), calculated using the dilute limit diffusion coefficient as an approximation of $D_s$ and the experimental $R_h$ (see "Methods").

Our experimental results indicate that as the protein concentration increases, there is a continuous transition from simple Brownian motion to anomalous diffusion. This transition is characterized by the deviation from single-exponential behavior in the intermediate scattering function, particularly evident at high protein concentrations, where two distinct decay processes become apparent. The hydrodynamic function, $H(q)$, is determined experimentally and serves as a signature of many-body hydrodynamic and direct protein interactions.

Modeling based on the $\delta\gamma$-theory of hydrodynamically interacting colloidal spheres in the short-time limit[42–44] shows quantitative agreement with the experimental results only by accounting for both short- and long-time diffusion.

## Results

### Scattering intensity $I(q)$ and structure factor $S(q)$

The experiments were conducted at the Material Imaging and Dynamics (MID) instrument of the EuXFEL[45], where the protein solutions, contained in capillaries, were measured using a small-angle X-ray scattering (SAXS) geometry with photon energy 10 keV (see "Methods"). Figure 2a shows the azimuthally integrated intensity, $I(q)$, for various protein concentrations, ranging from $c = 9$ mg/ml (volume fraction $\phi = 0.0042$) to $c = 730$ mg/ml ($\phi = 0.34$). A spherical form factor fit to the scattering intensity of the dilute solution ($c = 9$ mg/ml) yields a radius $R_s \approx 4.0$ nm (see also Supplementary Section 1), corresponding to the size of the ferritin cavity[46]. This result suggests that the SAXS signal predominantly originates from the protein core, which exhibits a higher scattering cross-section at this photon energy compared to the protein shell, because of the high iron content.

For the more concentrated solutions ($c = 70, 180, 400,$ and 730 mg/ml; see Table 1 for specific sample compositions), $I(q)$ exhibits correlation peaks that are indicative to the formation of a structure factor $S(q)$, as shown in Fig. 2b (see "Methods" for details on data analysis). The presence of the $S(q)$ peak at $q = q_0$ reflects interference between proteins, with a correlation length $\xi_p = 2\pi/q_0$. The values of $q_0$ and $\xi_p$ are provided in Table 2. As the protein concentration increases, the peak shifts to higher $q$-values, indicating that the inter-protein correlation length, $\xi_p$, decreases due to crowding. Additionally, Fig. 2b demonstrates that the $S(q)$ measured at the European Synchrotron Radiation Facility (ESRF), beamline TRUSAXS (ID02), is consistent with the $S(q)$ obtained at the EuXFEL.

### Intensity autocorrelation functions $g_2(q, t)$ and $q$-dependent diffusion coefficient $D(q)$

The intensity autocorrelation function, $g_2(q, t)$, is shown in Fig. 3a–d (see "Methods" for definition). The $g_2(q, t)$ was collected for varying concentrations ($c = 70, 180, 400,$ and 730 mg/ml) and $q$-values ($q = 0.225–0.625$ nm$^{-1}$), as indicated by the color-code. The solid lines represent the stretched exponential fits according to the equation

$$g_2(q, t) = 1 + \beta(q) \exp[-2(\tilde{\Gamma}(q)t)^{\alpha}], \tag{1}$$

where $\beta(q)$ is the speckle contrast, modeled as in ref. 36, $\tilde{\Gamma}(q)$ is the decorrelation rate and $\alpha$ is the Kohlrausch–Williams–Watts (KWW) exponent[47]. As protein concentration increases, we observe an overall slowdown of dynamics and a systematic change of the $g_2(q, t)$ lineshape from an almost simple exponential decay ($\alpha = 1$) at $c = 70$ mg/ml to a stretched exponential decay ($\alpha < 1$) at higher concentrations. The condition $\alpha < 1$ indicates heterogeneous protein motion at elevated concentrations. To account for the varying KWW exponents, we compute the average decorrelation rate[48] $\Gamma(q) = \tilde{\Gamma}(q)\alpha/\Gamma_f(1/\alpha)$, where $\Gamma_f(x)$ is the $\Gamma$-function.

The average decorrelation rate, $\Gamma(q)$, is shown in Fig. 4a. The solid lines denote the expected relationship $\Gamma(q) \propto q^2$ for Brownian diffusion. However, the experimental data deviate from this behavior, as highlighted by the shaded regions. To quantify these deviations, we calculate the $q$-dependent diffusion coefficient, $D(q) = \Gamma(q)/q^2$, shown in Fig. 4b. The diffusion coefficient $D(q)$ decreases as a function of $q$, reaching a minimum at $q = q_0^D$, which coincides with $q_0$ (see Table 2). This modulation in $D(q)$ is due to the De Gennes narrowing[49], observed in a variety of systems[50–55], including apoferritin[20,56,57]. Thus, the observed $D(q)$ indicates that collective protein motion is hindered (i.e., slowed down) by neighboring proteins, at length scales approaching the correlation length $\xi_p = 2\pi/q_0$.

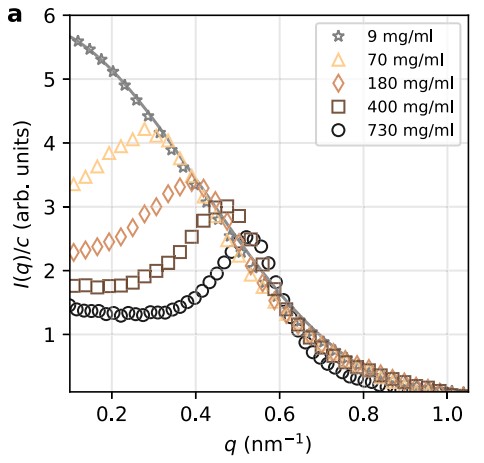

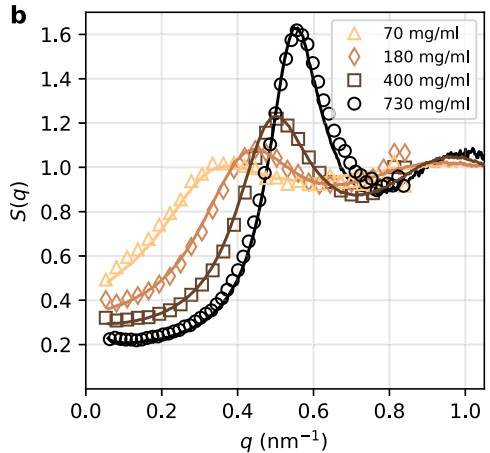

**Fig. 2 | Small-angle X-ray scattering (SAXS) of ferritin solutions. a** Scattering intensity as a function of momentum transfer, $I(q)$, for different protein concentrations ($c = 9$ mg/ml to $c = 730$ mg/ml) obtained at the European XFEL (EuXFEL). The $I(q)$ curves are background-subtracted and normalized by the respective concentration $c$. **b** The structure factor as a function of momentum transfer, $S(q)$, obtained from the experimental data collected at the EuXFEL (empty symbols) shows excellent agreement with the $S(q)$ obtained at the European Synchrotron Radiation Facility (ESRF, solid lines).

**Table 1 | Samples composition including protein concentration $c$, glycerol volume fraction $v_g$, and NaCl concentration $c_s$**

| c (mg/ml) | $v_{glyc}$ | $c_s$ (mM) |
|---|---|---|
| 70 | 0.55 | 15 |
| 180 | 0.55 | 35 |
| 400 | 0.55 | 80 |
| 730 | 0 | 150 |

### Hydrodynamic function $H(q)$ and self-diffusion coefficient $D_s$

In the short-time limit, the diffusion coefficient $D(q)$ can be linked to the hydrodynamic interactions by computing the hydrodynamic function $H(q)$[15], defined as:

$$H(q) = \frac{D(q)}{D_0} S(q), \tag{2}$$

where $D_0$ is the diffusion coefficient in the dilute limit and $S(q)$ the structure factor. Since there is no theoretical description beyond the short-time limit, we use Eq. (2) to estimate the hydrodynamic function, even though the time scale probed exceeds the interaction time and thus can also be influenced by long-time diffusion.

The hydrodynamic function $H(q)$ can be expressed as a sum of two contributions via[43]

$$H(q) = \tilde{H}(q) + H_0, \tag{3}$$

where $\tilde{H}(q)$ is the $q$-dependent term determined by protein–protein hydrodynamic interactions and the spatial arrangement of the proteins in the solution, i.e., the $S(q)$, while $H_0$ is a $q$-independent term, related to the protein self-diffusion $D_s$ by $H_0 = D_s/D_0$. At length scales much smaller than the typical protein–protein correlation length ($q \gg q_0$), the contribution of $\tilde{H}(q)$ is negligible and thus $H(q \gg q_0) = D_s/D_0$.

Figure 5a shows the experimental $D(q)S(q)/D_0$ for different ferritin concentrations ($c = 70, 180, 400,$ and $730$ mg/ml) using Eq. (2). Here, $S(q)$ was determined from the SAXS data (shown in Fig. 2b) and $D_0$ from the DLS data (see "Methods" and Supplementary Section 3). The value of $D_0$ was corrected to account for beam-induced heating (approximately $\Delta T \approx 1$ K as detailed in Table S1). $D(q)S(q)/D_0$ exhibits maxima at $q = q_0^H$ in agreement with the $q_0$ values listed in Table 2. At

increased protein concentrations, the overall magnitude of $D(q)S(q)/D_0$ decreases due to the reduction in self-diffusion $D_s$ and enhanced hydrodynamic interactions.

### Modeling based on the $\delta\gamma$-theory

$H(q)$ was modeled using the colloid $\delta\gamma$-theory[15,42,43,58]. This theory accounts for many-body hydrodynamic interactions of spherical particles in suspension in the short-time limit and uses $S(q)$ as input, as described in the "Methods" section. Figure 5a shows the results of the model (solid lines), including a scaling factor that was fitted to the experimental $D(q)S(q)/D_0$ data (see Supplementary Section 5). The model demonstrates good agreement with $D(q)S(q)/D_0$ within the measured $q$-range. The extracted $D_s/D_0$ as a function of the hydrodynamic volume fraction is shown in Fig. 5b (see also Fig. S7). The hydrodynamic volume fraction is $\phi_h = \phi R_h^3/R_p^3 \approx 1.59\phi$, where $R_h$ is the protein hydrodynamic radius ($R_h = 7.3$ nm obtained by DLS, see "Methods") and $R_p$ the protein radius ($R_p = 6.25$ nm based on the structure reported in ref. 59).

Alternatively, the hydrodynamic volume fraction $\phi_h$ dependence of $D_s/D_0$ can be modeled independently[60] by the expression

$$D_s^{short}(\phi_h)/D_0 = 1 - 1.73\phi_h + 0.88\phi_h^2, \tag{4}$$

as shown in Fig. 5b (dashed line). This formula, which assumes that $D_s$ refers to short-time self-diffusion, appears to overestimate $D_s/D_0$ compared to the experimental values extracted by fitting $D(q)S(q)/D_0$. Such deviations from colloid theory has been observed for various protein systems, including myoglobin[19], hemoglobin[21], and $\gamma_B$ crystalline[22]. In the case of hemoglobin[21], this discrepancy was partially explained by accounting for the hydrodynamic volume fraction. Here, implementing the $\phi_h$ correction is insufficient to explain the discrepancy seen for ferritin, where $\phi_h$ is included in Fig. 5. Another possible explanation for an overestimation of $D_s$ by the theory is the presence of patchy interactions (non-isotropic protein-specific interactions)[22], which is presumably *not* the case for ferritin given the repulsive interactions visible in the $S(q)$ lineshape, characterized by the reduction in $S(q = 0)$ with increasing concentration and the presence of a strong correlation peak[17,18,61].

At the observed time scales, it is possible that $D_s/D_0$ is influenced by a combination of short-time and long-time diffusion as previously observed by fluorescence recovery after photobleaching (FRAP) experiments[62,63]. To explore this hypothesis, the long-time self-

**Table 2 | Peak positions $q_0$ for various protein concentrations $c$ and corresponding volume fractions $\phi$**

| c (mg/ml) | $\phi$ | S(q) | D(q) | H(q) |
|---|---|---|---|---|
| 70 | 0.03 | $q_0 = 0.36$ nm$^{-1}$, $\xi_p = 17.2$ nm | $q_0^D = 0.36$ nm$^{-1}$ | $q_0^H = 0.38$ nm$^{-1}$ |
| 180 | 0.08 | $q_0 = 0.45$ nm$^{-1}$, $\xi_p = 14.0$ nm | $q_0^D = 0.44$ nm$^{-1}$ | $q_0^H = 0.45$ nm$^{-1}$ |
| 400 | 0.19 | $q_0 = 0.50$ nm$^{-1}$, $\xi_p = 12.6$ nm | $q_0^D = 0.49$ nm$^{-1}$ | $q_0^H = 0.49$ nm$^{-1}$ |
| 730 | 0.34 | $q_0 = 0.55$ nm$^{-1}$, $\xi_p = 11.5$ nm | $q_0^D = 0.55$ nm$^{-1}$ | $q_0^H = 0.55$ nm$^{-1}$ |

The columns represent the peak positions for the structure factor $S(q)$ (including the correlation length $\xi_p = 2\pi/q_0$), the minimum in the diffusion coefficient $D(q)$ ($q_0^D$) and the peak position in the hydrodynamic function $H(q)$ ($q_0^H$), both estimated from the $\delta\gamma$-model. The error bars for the peak position estimation are 0.01 nm$^{-1}$, due to the $q$-resolution of the modeled data, and the $q$-binning of the experimental data.

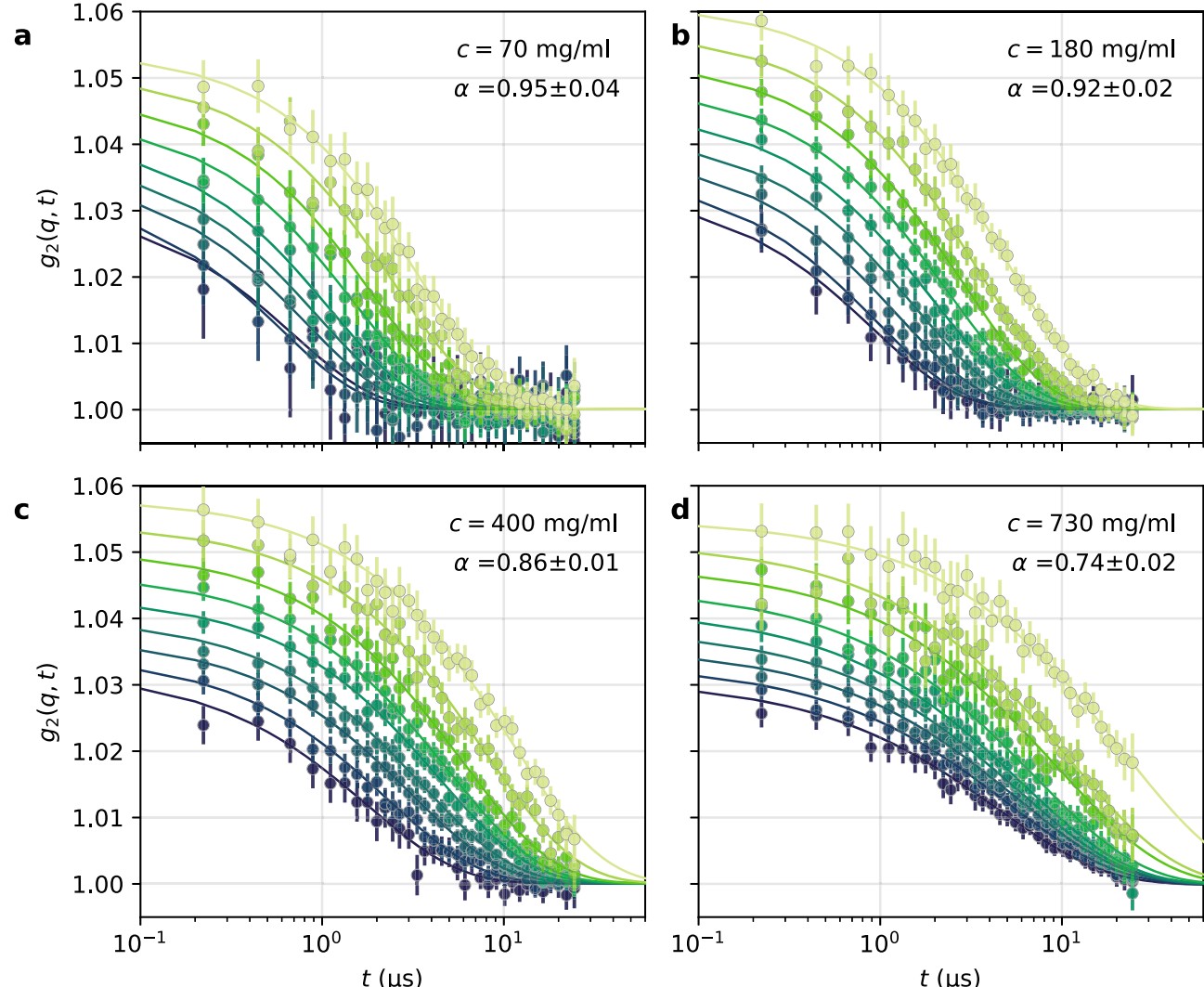

**Fig. 3 | X-ray photon correlation spectroscopy (XPCS) data of ferritin solutions obtained at EuXFEL.** The intensity autocorrelation function, $g_2(q, t)$, for different protein concentrations **a** $c = 70$ mg/ml, **b** $c = 180$ mg/ml, **c** $c = 400$ mg/ml, and **d** $c = 730$ mg/ml. Data in panels (**a**–**c**) were measured in water-glycerol (with glycerol volume fraction $v_{glyc} = 0.55$), while panel (**d**) presents data measured in water to reach the desired protein concentration ($c = 730$ mg/ml). The different colors represent different $q$-values, changing from lighter to darker green for $q$ increasing from $q = 0.225$ nm$^{-1}$ to $q = 0.625$ nm$^{-1}$ with equal spacing of $dq = 0.05$ nm$^{-1}$. Solid lines represent stretched exponential fits, with the corresponding Kohlrausch–Williams–Watts (KWW) exponent $\alpha$ shown in the legend. The error bars on the KWW exponents are estimated from the stretched exponential fit. The error bars on $g_2(q, t)$ shown correspond to the standard error, estimated as described in the SI.

diffusion coefficient is approximated by[63,64]

$$D_s^{long} = \frac{D_s^{short}}{D_0} \cdot D_s^{direct} \tag{5}$$

The short-time self-diffusion coefficient, $D_s^{short}$, contains the hydrodynamic interactions, but not the direct protein interactions (i.e., electrostatic and nonspecific attractive interactions). The latter are included in $D_s^{direct} = D_0/(1 + 2\phi_h\chi)$, where $\chi$ is the contact value of the pair-correlation function[65], which was

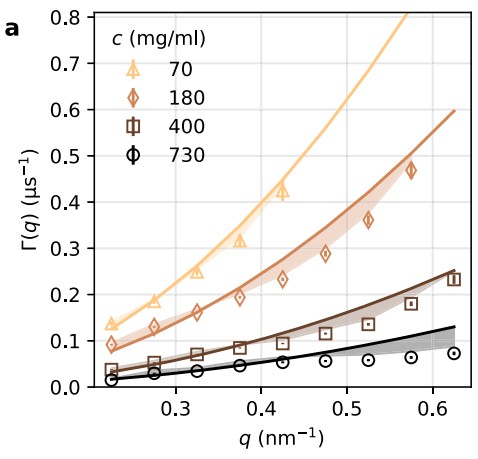

**Fig. 4 | Average decorrelation rate Γ(q) and diffusion coefficient D(q) at different protein concentrations. a** Decorrelation rate Γ(q) as a function of momentum transfer q. Solid lines represent fits with $\Gamma(q) = Dq^2$, while shaded areas highlight deviations of the experimental values from the fit. **b** Diffusion coefficient

$D(q) = \Gamma(q)/q^2$ as a function of momentum transfer q. Solid lines are model fits using δγ-theory. The error bars are estimated from the stretched exponential fit including standard error propagation.

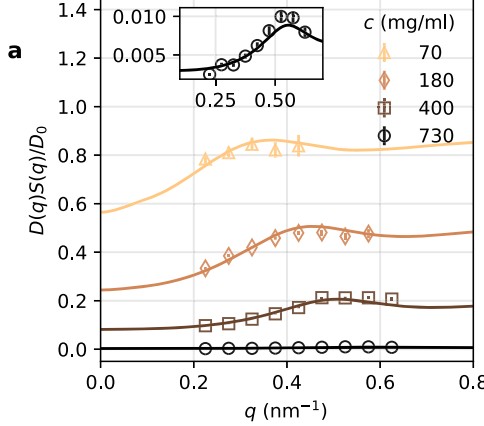
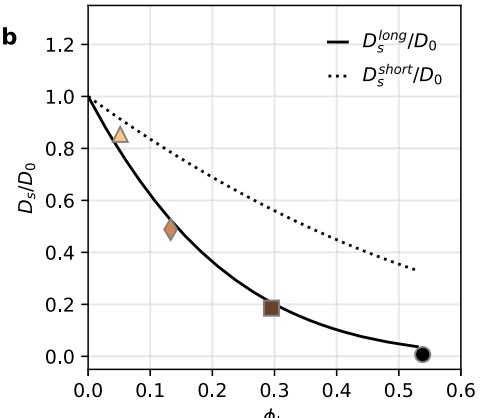

**Fig. 5 | $D(q)S(q)/D_0$ and self-diffusion coefficient $D_s$ for different concentrations. a** $D(q)S(q)/D_0$ as a function of momentum transfer q (empty symbols) and model results using the δγ-theory (solid lines). The inset displays a zoom in for c = 730 mg/ml. The error bars are estimated from the stretched exponential fit including standard error propagation. **b** The ratio of the self-diffusion coefficients over the dilute-limit diffusion coefficients, $D_s/D_0$, as a

function of the hydrodynamic volume fraction $\phi_h = \phi \frac{R_h^3}{R_p^3}$. The lines represent the expected volume fraction dependence of the short-time self-diffusion coefficient $D_s^{short}/D_0$[42] (dotted line) and the long-time self-diffusion coefficient $D_s^{long}/D_0$[62] (solid line). The error bars are estimated from the fit of the scaling factor including standard error propagation and, if not visible, they are smaller than the symbol.

estimated approximating the protein interactions as hard spheres (where $\chi = (1 + 0.5\phi_h)/(1 - \phi_h)^2$). This model appears to better describe the volume fraction dependence of $D_s/D_0$, as shown in Fig. 5b (solid line). Therefore, the experimental data indicate that the steep decrease of $D_s/D_0$ with increasing $\phi_h$ is likely due to the $D_s^{long}$ component, which becomes dominant with increasing volume fraction.

**Cage effects and long-time diffusion**

To investigate the origin of the heterogeneous dynamics indicated by $\alpha < 1$ for high protein concentrations, we calculate the intermediate scattering function, $f(q, t)$, related to the $g_2(q, t)$ by the Siegert relation,

$$g_2(q, t) = \beta(q)|f(q, t)|^2 + 1. \tag{6}$$

For Brownian diffusion in the dilute limit, $f(q, t)$ is expected to be a simple exponential function, meaning that $\ln[f(q, t)]$ depends linearly on time, and $\ln[f(q, t)] = -\langle r^2(t)\rangle q^2$ with $\langle r^2(t)\rangle$ the mean square

displacement. Figure 6a shows $f(q, t)$ for different protein concentrations, where the dotted line corresponds to a linear fit over the first 5 μs. We observe that for c = 70 mg/ml, the data follow closely the single exponential fit, while deviations from single exponential behavior become apparent at higher concentrations. Notably, at c = 730 mg/ml, a distinct slope is observed at time scales longer than 5 μs. This deviation is evident for all measured q values, as shown in Fig. 6b, and becomes more pronounced when $\ln[f(q, t)]$ is normalized by the slope of the linear fit over the first 2.5 μs, as seen in Fig. 6d. This observation suggests the presence of two components in the decay.

Using a double exponential fit, we extract the diffusion coefficients of the two components, by:

$$g_2(q, t) = 1 + \beta(q)\{[1 - A(q)]\exp[-\Gamma_1(q)t] + A(q)\exp[-\Gamma_2(q)t]\}^2, \tag{7}$$

where $\Gamma_1(q)$ and $\Gamma_2(q)$ correspond to fast and slow decorrelation rates and $A(q)$ represents the relative amplitude of the slow component. Figure 6c shows the diffusion coefficients, $D^{short}(q) = \Gamma_1(q)/q^2$

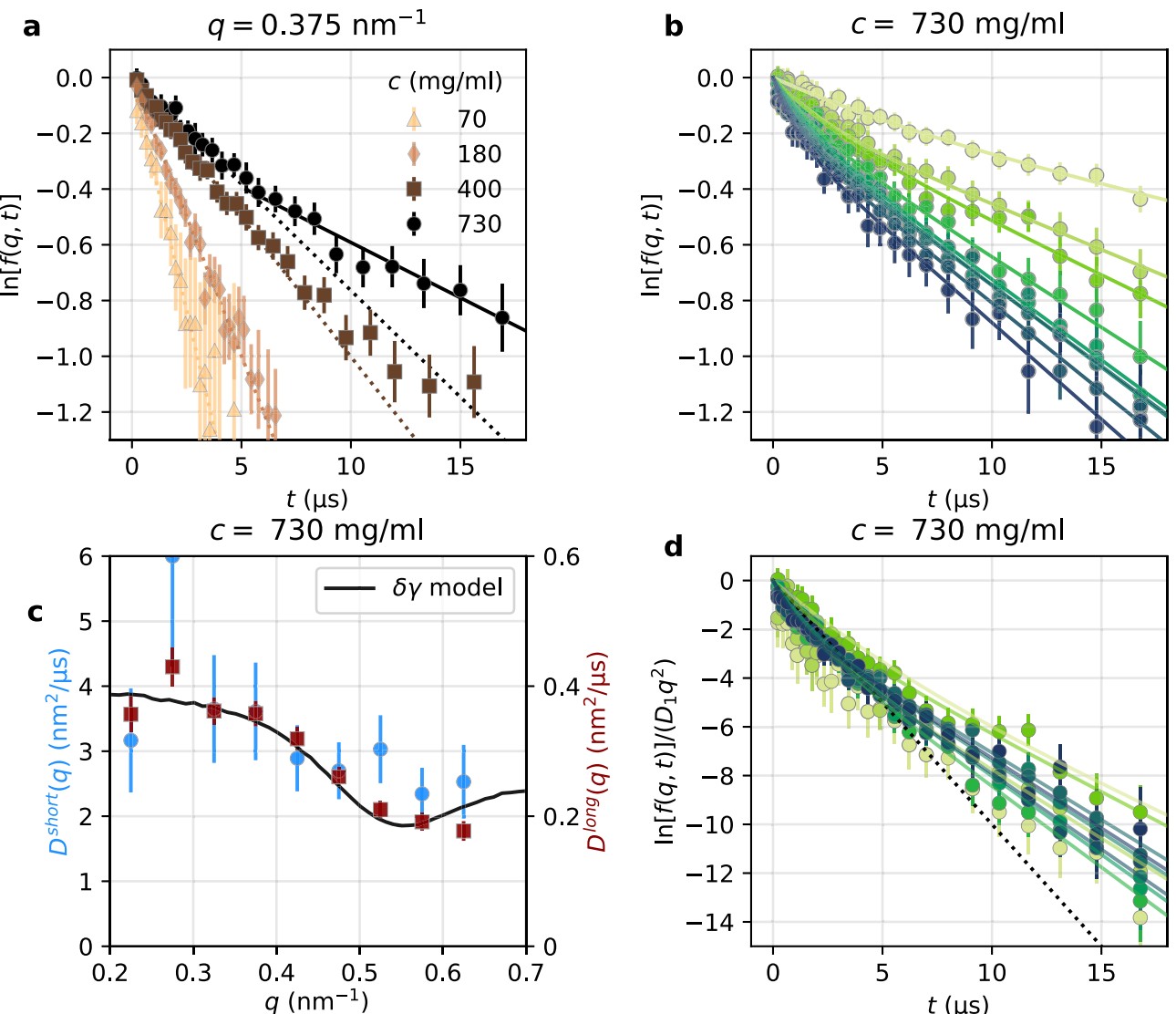

**Fig. 6 | Evidence of short- and long-time diffusion. a** The intermediate scattering function $f(q, t)$ as a function of time (shown in $\ln[f(q, t)]$) at $q = 0.38 \ nm^{-1}$ for different ferritin concentrations ($c = 70, 180, 400, 730 \ mg/ml$). The dotted lines are linear fits in the time range up to 5 μs. The solid line is a linear fit for time longer than 5 μs. **b** The $\ln[f(q, t)]$ of ferritin $c = 730 \ mg/ml$ at $q$ values from $q = 0.225 \ nm^{-1}$ to $q = 0.625 \ nm^{-1}$ from lighter to darker green. For (**a**) and (**b**), the errors were estimated through error propagation from the estimated error of $g_2(q, t)$. **c** The $q$-dependent diffusion constants extracted with a double exponential fit. The blue circles are

$D^{short}(q) = \Gamma_1(q)/q^2$ (left hand side y-axis) and the red squares are $D^{long}(q) = \Gamma_2(q)/q^2$ (right hand side y-axis). The solid black line is the modeled $D(q)$, from the $\delta\gamma$-theory also shown in Fig. 4b and refers to the y axis on the right hand side. The error bars correspond to error propagation from the fit parameters and, if not visible, they are smaller than the symbol. **d** $\ln[f(q, t)]$ normalized by the slope of $\ln[f(q, t)]$ in the time range up to 2.5 μs. The dotted line is a linear fit in the time range up to 5 μs. The error bars were obtained with error propagation from the error on $\ln[f(q, t)]$.

and $D^{long}(q) = \Gamma_2(q)/q^2$, obtained from the decorrelation rates of the double exponential fit. $D^{short}(q)$ and $D^{long}(q)$ exhibit similar $q$-dependence and minima for $q = q_0$ in agreement with the behavior of $D(q)$ at lower concentrations, shown in Fig. 4. At $c = 730 \ mg/ml$, the ratio $D^{long}(q)/D^{short}(q)$ is 0.09 ± 0.02.

One possibility is that the slow component is due to aggregates with size $\approx 11.1$ times larger than a single protein (based on $D^{short}(q)/D^{long}(q)$), predicting a correlation peak at $q = q_0/11.1 = 0.050 \ nm^{-1}$. However, in the $S(q)$ data (Fig. 2b) we do not see any enhancement at low $q$, which would indicate such a peak. Alternatively, the two decays represent short- and long-time diffusion. From $D^{short}(q)$, we estimate $D_s^{short} = D^{short}(q \rightarrow \infty) = 2.35 \ nm^2 \mu s^{-1}$ which yields an interaction time of $\tau_i = R_h^2/(6D_s^{short}) = 3.77$ μs. This value corresponds to the cross-over time between the two exponential functions, supporting the

hypothesis that the double exponential decay arises from short- and long-time diffusion. It is noteworthy, that estimating the interaction time based on the Stokes-Einstein equation in the dilute limit for ferritin in water yields $\tau_i \approx 0.3$ μs, whereas in crowded conditions $\tau_i$ is $\approx 12.6$ times larger. This calculation implies a significantly slower reaction rates than those predicted from simple Stokes-Einstein estimations, even at molecular length scales.

The short-time diffusion coefficient $D^{short}(q)$ is almost 15 times smaller than the dilute-regime diffusion coefficient. The reason for the discrepancy between $D^{short}(q)$ and the dotted line in Fig. 5b is due to the limit of the validity of Eq. (4) to volume fractions below 0.45 according to ref. 60. For ferritin in heavy water at $\phi = 0.153$ and $\phi = 0.327$ (corresponding to $\phi_h$ of 0.21 and 0.44), the values of the normalized short-time self-diffusion $D_s/D_0$, extracted with simulations and experimental

data employing neutron scattering[20], are 0.76 and 0.48, respectively. These values are close to the expected short-time diffusion coefficient (dotted line in Fig. 5b). Hence, we conclude that the large difference seen between the samples in glycerol-water mixtures and water cannot be attributed to the different solvents, but rather to very elevated crowding conditions of the proteins in solution at 730 mg/ml.

The lineshape of $D^{short}(q)$ and $D^{long}(q)$ coincides with the prediction of the $\delta\gamma$-theory, as shown in Fig. 6c (black line). Since hydrodynamic forces influence both diffusion coefficients, while direct interactions are relevant only in the long-time diffusion, we conclude that hydrodynamic forces act similarly on the short- and long-time diffusion and the direct interactions do not provide additional $q$-dependent changes on the diffusion coefficient.

These results are also in agreement with previous studies performed on larger colloidal particles[66–68], where the short- and long-time diffusion coefficients, $D^{short}(q)$ and $D^{long}(q)$, were shown to have the same lineshape at $qR > 2.5$. This finding indicates that for the investigated system at $1.5 < qR < 4$, the direct interactions, relevant only for the long-time diffusion, do not influence the $q$-dependence of $D^{long}(q)$. Consequently, hydrodynamic functions can be effectively extracted from long-time diffusion, disentangling the contributions of direct forces and hydrodynamic interactions (see also Supplementary Section 5).

Additionally, the relative amplitude of the two decays, $A(q)$, provides insight into protein motion. It is given by:

$$A(q) = A_0 \exp\left(-\frac{q^2\delta^2}{6}\right), \tag{8}$$

reflecting the loss of correlation due to a Debye-Waller-like movement around a central position, with average displacement $\delta$ and percentage $A_0$ of particles involved[69]. For this system, $A_0 = 89 \pm 3\%$ and $\delta = 1.0 \pm 0.3$ nm indicating that the majority of protein molecules participate in cage formation, with an average displacement of 1.0 nm. This finding suggests that protein rattling within the cage occurs only within a fraction of their radius due to the limited available space at high concentrations. This result aligns with the expected reduction in cage size due to confinement with increasing volume fraction, approaching the glass transition[12,66,70]. Cage effects have been probed also in biological systems[28,71] and inorganic colloids with different sizes[66]. Here, the main parameter that drives the cage formation is the excluded volume effects, which can be affected by the inter-particle interactions[72].

Here, we have used two models to extract the diffusion coefficient: the stretched exponential and the double exponential models. In the stretched exponential the short- and long-time diffusion are effectively merged into a single time constant, and the exponent indirectly reflects the extent to which these two components are separated as a function of concentration. In the double exponential model, the long- and short-time diffusion are treated separately and the amplitude indicates the relative contributions of the two components. The latter indicates that 90% of the decay is from the long-time diffusion component. Since the contribution of long-time diffusion component is predominant, the extracted value of $D(q)$ obtained from the stretched exponential model is effectively close to the value of $D^{long}(q)$. The solid black line shown in Fig. 4b and in Fig. 6c (it refers to the y-axis on right-hand side) is the same between the same panels, showing good agreement between $D(q)$ and $D^{long}(q)$.

## Discussion
In summary, we present results from MHz-XPCS experiments conducted at the EuXFEL, complemented by modeling based on colloid theory, focusing on concentrated ferritin solutions. A key advantage of this experimental approach is its ability to capture long-time protein diffusion at length scales comparable to the average protein–protein distance in crowded conditions. In contrast to short-time diffusion, there are no theoretical predictions for long-time diffusion that adequately account for hydrodynamic interactions and direct interactions.

Our results reveal anomalous diffusion in highly concentrated ferritin solutions, which becomes more pronounced with increasing protein concentration. From the momentum-transfer dependent diffusion coefficient, $D(q)$, and the corresponding structure factor, $S(q)$, we extract the hydrodynamic function $H(q)$, which quantifies contributions from many-body hydrodynamic interactions. Modeling based on the $\delta\gamma$-theory of hydrodynamically interacting colloidal spheres[42,43] shows good agreement with the experimental $H(q)$ when a scaling is applied. Notably, the $H(q)$ intercept, related to the ratio of the self- over the dilute-limit diffusion coefficients, $D_s/D_0$, decreases with volume fraction much more rapidly than expected for purely short-time diffusive motion. This finding indicates that both short- and long-time diffusion contribute to self-diffusion, with the long-time diffusion becoming more significant as the protein concentration increases.

Further evidence for the coexistence of short- and long-time diffusion comes from the analysis of the intensity autocorrelation function, $g_2(q, t)$. The $g_2(q, t)$ exhibits a double exponential decay with increasing protein concentration, which is especially pronounced at the highest concentration, $c = 730$ mg/ml. The data suggest that the short- and long-time diffusion coefficients, $D^{short}(q)$ and $D^{long}(q)$, exhibit nearly identical $q$-dependence differing only by a scaling factor, as previously observed in colloidal suspensions[66]. This finding indicates that long-time diffusion is governed by both hydrodynamic effects and direct interactions. We observe that the latter contribute to an $q$-independent decrease of the diffusion coefficient for $1.5 < qR < 4$. At the highest concentration ($c = 730$ mg/ml), cage effects are observed with an average displacement of $\delta = 1.0 \pm 0.3$ nm, indicating that cage-trapping occurs within a fraction of the protein radius due to confinement. These effects can quantitatively explain the observed anomalous diffusion, but also provide a molecular mechanism for the reduced mobility of proteins under crowded conditions, and is consistent with simulations on globular proteins[72].

From a broader perspective, our findings can facilitate biomedical applications involving ferritin as a nano-drug delivery system[39] and nano-reactor[73], as in sustained-release preparations the accurate determination of the anomalous diffusion coefficients can significantly influence the bioavailability profile. Furthermore, ferritin has been suggested as an MRI contrast agent given the paramagnetic properties of its core[74], and the possibility of exchanging the content of the core[75]. In this context, ferritin aggregation may be a useful strategy to produce a functional reporter gene for magnetic resonance imaging[76]. Future work that builds on the results presented here has the potential to focus on environments that more closely resembles biological conditions, by introducing crowders of different sizes (e.g., polymers, proteins, polysaccharides) in order to explore the effects of polydispersity on long-time diffusion[77].

## Methods
### Sample preparation
Protein solutions of different concentrations were prepared in steps, starting from centrifuging the parent protein saline solution and followed by dilution with glycerol. Ferritin from equine spleen solutions (Sigma-Aldrich, F4503) was used as the parent solution, with an initial protein concentration of $c = 71$ mg/ml and NaCl content of 150 mM. This parent solution was centrifuged at $10,000 \times g$ for 1 h using 10 kDa Millipore filters, resulting in a concentrated ferritin solution of $c = 730 \pm 20$ mg/ml, as determined by UV-Vis spectroscopy at $\lambda = 280$ nm[78], where the error bar corresponds to the standard error. This concentration was estimated using a calibration curve based on samples of known concentration.

**Table 3 | The X-ray parameters used at the European XFEL (EuXFEL) and European Synchrotron Radiation Facility (ESRF)**

| Parameter | EuXFEL (MID) | ESRF (IDO2) |
|---|---|---|
| Photon energy (keV) | 10.0 | 12.23 |
| Frame rate | 4.5 MHz/10 Hz (intra-/inter-train) | 10 Hz |
| Sample environment | 1mm quartz capillaries | 1 mm quartz capillaries |
| Detector | AGIPD | EIGER2 4M |
| Pixel size (µm²) | 200 × 200 | 75 × 75 |
| Sample-detector distance (m) | 7.687 | 1.48 |

The concentrated ferritin solution was subsequently diluted with glycerol and water to achieve protein concentrations $c = 70, 180$, and 400 mg/ml (with 55 vol% glycerol). As the protein concentration decreased, the NaCl concentration also decreased, as the diluting solvent did not contain NaCl. These solutions were filled in quartz capillaries with an outer diameter of $\approx 1$ mm and a wall thickness of $\approx 20$ µm. The volume fraction is calculated from the protein concentration, $c$, using the equation $\phi = cv_s$, where $v_s = 0.463$ ml/g is the specific volume of the protein[79], which is the inverse of the protein density. A summary of the samples and their composition is shown in Table 1.

### X-ray experimental parameters
The data were acquired at the MID instrument[45] of the EuXFEL in small-angle X-ray scattering (SAXS) configuration using a pink beam with a photon energy of 10 keV. The X-ray pulses, with an intra-train repetition rate of 4.5 MHz (222.2 ns pulse spacing), were delivered in trains of up to 310 pulses, with an inter-train repetition rate of 10 Hz. The Adaptive Gain Integrating Pixel Detector (AGIPD)[80,81] was positioned at a sample-detector distance of 7.687 m. The X-ray beam was focused to a diameter of $12 \pm 1$ µm and the X-ray bandwidth was $\Delta E/E = 7 \times 10^{-3}$, determined by fitting the speckle contrast. This beam size, achieved using compound refractive lenses (CRL), was chosen to enhance both the speckle contrast and the signal-to-noise ratio (SNR) of the XPCS measurements[82]. X-ray intensity was adjusted by chemically vapor deposited (CVD) diamond attenuators of variable thickness to minimize beam-induced effects while maintaining high SNR (see Table S1). To refresh the sample volume between trains, the sample was moved through the X-ray beam at a translation motor speed of 0.4 mm s⁻¹, as described in ref. 36. Table 3 summarizes the experimental parameters for the EuXFEL measurements, along with the SAXS experimental parameters at ESRF (beamline ID02). The measurements were repeated to increase the SNR. This also ensured that the sample was measured in different positions along the capillary to take into account local variations, which were not seen. The total repetition number for the data shown is 6000 for each protein concentration for an X-ray transmission of $5.35 \times 10^{-6}$.

### Brownian diffusion coefficient $D_0$ and hydrodynamic radius $R_h$
The protein diffusion coefficient, $D_0$, was measured in dilute conditions ($c = 9$ mg/ml) using DLS (see Section 4 of the Supplementary Information). For ferritin in water-glycerol (55 vol% glycerol), we obtained $D_0 = 3.05 \pm 0.01$ nm² µs⁻¹. For ferritin in water, the diffusion coefficient used is $D_0 = 30.1 \pm 0.2$ nm² µs⁻¹. The hydrodynamic radius, $R_h$, was calculated using the Stokes-Einstein equation, $D_0 = k_B T/(6\pi\eta R_h)$, where $\eta$ is the viscosity, $T$ the temperature and $k_B$ the Boltzmann constant. Here, for water-glycerol by using $\eta = 9.3 \pm 0.6$ mPa s[83] and $T = 298$ K, we estimated $R_h = 7.3 \pm 0.1$ nm. This value is comparable to literature values of apoferritin (hollow shell without the core) in saline solutions obtained with DLS ($R_h = 6.9$ nm[17], $R_h = 6.3$ nm[18]) and HYDROPRO ($R_h = 6.4$ nm)[84]. The slightly larger $R_h$ obtained here could

be due to the presence of the iron (ferritin vs apoferritin), which may influence the protein hydrodynamic radius as surrounding hydration layer. For ferritin solutions, the $R_h = 6.85 \pm 0.8$ nm obtained by DLS[85] is consistent with our estimate.

### SAXS data analysis
The protein scattering intensity, $I(q)$, was determined by subtracting the background scattering intensity, $I_{bkg}$ (scattering from the capillary containing only the solvent), from the measured scattering intensity, $I_m(q)$:

$$I(q) = I_m(q) - I_{bkg}(q). \tag{9}$$

The scattering intensity, $I(q)$, of mono-dispersed spherical proteins is modeled by

$$I(q) = c\Delta\rho^2 P(q)S(q), \tag{10}$$

where $\Delta\rho$ is the scattering contrast between the protein and the solvent, $c$ the protein concentration, $S(q)$ the structure factor, and $P(q)$ the form factor. The $P(q)$ was determined experimentally by measuring the scattering intensity in the dilute limit, $I_0(q)$, where $S(q) \approx 1$. Here, $I_0(q)$ is estimated using low concentration sample ($c = 9$ mg/ml). The experimental $S(q)$ was then calculated as follows:

$$S(q) = C\frac{I_m(q) - I_{bkg}(q)}{I_0(q) - I_{bkg}(q)}, \tag{11}$$

where $C$ is a normalization constant. More detailed are provided in section 1 of the Supplementary Information.

### Intensity autocorrelation function $g_2$ calculation and fit
The $g_2(q, t)$ function was calculated starting from the two time correlation function (TTCs), which represent the correlation between speckle images taken at times $t_1$ and $t_2$ at momentum transfer $q$:

$$c_2(q, t_1, t_2) = \frac{\langle I(q, t_1)I(q, t_2)\rangle}{\langle I(q, t_1)\rangle\langle I(q, t_2)\rangle} \tag{12}$$

where $\langle ... \rangle$ indicates averaging over all pixels with the same momentum transfer, $q$. The TTCs functions were extracted with the MID data processing pipeline embedded into the DAMNIT tool of the EuXFEL data analysis group (https://damnit.readthedocs.io/en/latest/), which includes a correction using train cross-correlation[86]. The average TTC was obtained by averaging the TTCs of the single trains, where the weights correspond to the inverse of the standard errors on the TTCs squared. The corresponding error bars $\delta c_2(q, t_1, t_2)$ were estimated from the standard error between the individual TTCs. A baseline subtraction on the average TTCs was performed at each $q$. The value of the baseline was calculated as weighted mean of the delay times above 46 µs which correspond to pulse numbers from 210 to 310. Finally, $g_2(t, q)$ was obtained by $g_2(q, t) = \langle c_2(q, t_1, t_1 + t)\rangle_{t_1}$, with $\langle \cdot \rangle_{t_1}$ the weighted average, with weights corresponding to $1/\delta c_2(q, t_1, t_2)^2$. The error bars on $g_2(t, q)$, were estimated from the error transposition of the weighted average.

The fits of the exponential functions to $g_2(q, t)$ were performed simultaneously in both $q$ and $t$ dimensions, to ensure better stability of the fits. The fit contains the $q$-dependence and the overall values of the contrast $\beta(q)$ as described in Supplementary Section 6. The contrast $\beta(q)$ was included to overcome the limitation of 222.2 ns for the shortest time delay. The experimental parameters for the contrast estimation (X-ray bandwidth, sample-detector distance, photon energy, sample thickness, etc.) were fixed with the values reported in Table 3 except for the beam size. The latter was set as fit parameter to account for the inherent fluctuations of the XFEL beam (for values, see Supplementary

Section 6). The value of $\alpha$ was set to be constant with $q$. The $q$-dependence of $\bar{\Gamma}(q)$ was not fixed to be able to extract $D(q)$.

## Beam-induced effects assessment

Identifying the onset of X-ray beam-induced effects is crucial for MHz-XPCS. Beam-induced effects can manifest as fluence-dependent changes in the scattering intensity, $I(q)$, and the diffusion coefficient, $D(q)$. These changes may result from beam-induced heating or radicals formation in the solution, potentially leading to protein aggregation or denaturation[36].

Supplementary Fig. S2 shows the $I(q)$ for varying pulse accumulations at different concentrations, $c$, and transmissions, $T_0$. In addition, the $I(q)$ as a function of pulse number for $q = 0.2\,nm^{-1}$ is shown in Supplementary Fig. S3. Minor changes in the $I(q)$ are observed across all $T_0$. Specifically for data used for the figures presented, i.e., at $T_0 = 5.35 \times 10^{-6}$, intensity changes are less than 1%.

The fluence-dependence of $D(q)$ and TTCs are discussed in Supplementary Section 3. Within the error bars, changes in the KWW exponent $\alpha$ with dose are negligible. However, a slight increase in the diffusion coefficient is observed with increasing pulse number, likely due to beam-induced temperature rise[36,87], estimated to be below 1 K. This approach allows us to identify the optimal parameters to mitigate beam-induced effects, such as optimal transmission $T_0$, number of repetitions $n_{rep}$, and the number of pulses-per-train $n_{pulses}$. Considering the minor changes in both $I(q)$ and $D(q)$ under these conditions, we conclude that protein aggregation is not detected. The intrinsic protein dynamics can be accurately accessed by including a correction for the temperature increase (in the order of 1 K) as reported in Table S2.

## Hydrodynamic function modeling

The hydrodynamic function, $H(q)$, was modeled according to the framework outlined in refs. 43,58 by:

$$H(q) = \tilde{H}(q) + \frac{D_s(\phi)}{D_0} \tag{13}$$

The $q$-dependent part $\tilde{H}(q)$ is calculated using *jscatter*[88] with the expression:

$$\tilde{H}(q) = \frac{3}{2\pi} \int_0^\infty dR_p k \frac{\sin^2(R_p k)}{(R_p k)^2 [1 + \phi S_\gamma(R_p k)]} \int_{-1}^{1} dx(1-x^2)[S(|\mathbf{q}-\mathbf{k}|)-1], \tag{14}$$

where $x = \cos(\mathbf{q}, \mathbf{k})$ is the angle between $\mathbf{q}$ and $\mathbf{k}$ vectors, $\phi$ is the protein volume fraction. The protein radius $R_p = 6.25\,nm$ is based on the structure reported in ref. 59, and $S_\gamma(q)$ is the volume fraction dependent function given in ref. 58. The $S(q)$ used for the calculation is obtained by interpolating the experimental $S(q)$ (see Supplementary Section 1).

The normalized self-diffusion coefficient $\frac{D_s(\phi)}{D_0}$ was estimated via[58]:

$$\frac{D_s(\phi)}{D_0} = \frac{1}{2\pi} \int_0^\infty dR_p k \frac{\sin^2(R_p k)}{(R_p k)^2 [1 + \phi S_\gamma(R_p k)]} \tag{15}$$

More information on the fit is provided in Supplementary Section 5.

## Reporting summary

Further information on research design is available in the Nature Portfolio Reporting Summary linked to this article.

## Data availability

The raw data generated in this study have been deposited in the XFEL and ESRF databases under [doi:10.22003/XFEL.EU-DATA-003094-00] and [doi:10.22003/XFEL.EU-DATA-005397-00]. The data shown in the figures are provided in the Supplementary Information/Source data file. Source data are provided with this paper.

## Code availability

These data were analyzed using the DAMNIT tool of the EuXFEL data analysis group ([https://damnit.readthedocs.io/en/latest/]) described in ref. 89. The code used to perform the fitting and plotting routines shown in this study is available from the corresponding authors upon request.

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

## Acknowledgements

We acknowledge the European XFEL in Schenefeld, Germany, for provision of XFEL beamtime at the Scientific Instrument MID (Materials Imaging and Dynamics) and thank the staff for their assistance. The data presented here are collected as part of the measurement time awarded to proposals 3094 and 5397. In addition, we acknowledge Maxwell cluster for providing computer resources to perform the analysis. We thank the European Synchrotron Radiation Facility (ESRF) for provision of synchrotron radiation facilities at TRUSAXS instrument (Time Resolved Ultra Small Angle X-ray Scattering), IDO2 (SC-5485) with additional support from Theyencheri Narayanan. F.P. acknowledges financial support by the Swedish National Research Council (Vetenskapsrådet) under Grant No. 2019-05542, 2023-05339 and within the Röntgen-Ångström Cluster Grant No. 2019-06075, and the kind financial support from Knut och Alice Wallenberg foundation (WAF, Grant No. 2023.0052). This research is supported by the Center of Molecular Water Science (CMWS) of DESY in an Early Science Project, the Max-Water initiative of the Max-Planck-Gesellschaft, Carl Tryggers (Project No. CTS 21:1589) and the Wenner-Gren Foundations (Project No. UPD2021-0144). F.P., I.A., and A.G. acknowledge funding from the European Union's Horizon Europe research and innovation program under the Marie Skłodowska-Curie grant agreement No. 101081419 (PRISMAS) (F.P. and I.A.) and 101149230 (CRYSTAL-X) (F.P. and A.G.). We also acknowledge BMBR ErUM-Pro funding (05K22PS1, C.G.), BMBF (05K19PS1 and 05K20PSA, C.G.; 05K19VTB, F.S. and F.Z.), DFG-ANR (SCHR700/28-1, SCHR700/42-1, ANR-21-CE06-0047 IDPXN, F.S., F.Z., and T.S.). F.L. and P.P.R. thank the Cluster of Excellence "Advanced Imaging of Matter" of the Deutsche Forschungsgemeinschaft (DFG)-EXC 2056-project id no. 390715994. A.L. and C.G. acknowledge financial support by the consortium DAPHNE4NFDI in association with the National Research Data Infrastructure Germany (NFDI) e.V. - project number 46024879. M.P. thanks the DELTA machine group for providing synchrotron radiation for sample characterization.

## Author contributions

A.G., M.R., and F.P. designed the experiment, along with discussions with C.G., F.S., F.Z., M.P., J.M., T.S., N.D.A., and F.L. A.G., M.B., and M.F. prepared and handled the samples. J.H., A.R.-F., J.-E.P., F.B., U.B., W.L., W.J., M.Y., R.S., A.Z., J.M., and A.M. operated MID and collected data together with the rest of the experimental team. A.G., M.B., M.F., S.T., and I.A. performed online data processing and analysis at MID, based on scripts developed by M.R. and the MID data analysis team (JW and AL being the main contributors). N.D.A., S.B., S.R., M.D.S., M.K., J.S., L.F.R., A.M.R., M.S.A., C.H.W., P.P.R., and M.D. were in addition responsible for the elog. Experiments at ESRF were supported by W.C. A.G. performed offline data processing and analysis with input from F.P., F.Z., C.G., T.G., N.D.A., and M.P. All authors jointly performed the experiments and discussed the final results. The manuscript was written by A.G. and F.P. with input from all authors.

## Funding

## Competing interests

The authors declare no competing interests.

## Additional information

[1]Department of Physics, AlbaNova University Center, Stockholm University, Stockholm, Sweden. [2]Institut für Angewandte Physik, Universität Tübingen, Tübingen, Germany. [3]Department Physik, Universität Siegen, Siegen, Germany. [4]European X-Ray Free-Electron Laser Facility, Schenefeld, Germany. [5]Fakultät Physik/DELTA, TU Dortmund, Dortmund, Germany. [6]The Hamburg Centre for Ultrafast Imaging, Hamburg, Germany. [7]Institute for Biochemistry and Molecular Biology, Laboratory for Structural Biology of Infection and Inflammation, University of Hamburg, Hamburg, Germany. [8]ESRF - The European Synchrotron, Grenoble, France. [9]Institut Laue-Langevin, Grenoble, France. [10]Deutsches Elektronen-Synchrotron DESY, Hamburg, Germany.
✉e-mail: anita.girelli@fysik.su.se; f.perakis@fysik.su.se

