## [Transparent Peer Review file · Nature Communications]

Coherent X-rays reveal anomalous molecular diffusion and cage effects in crowded protein solutions

Corresponding Author: Dr Anita Girelli

Version 0:

Reviewer comments:

Reviewer #1

(Remarks to the Author)

This manuscript explores protein diffusion dynamics in a crowded environment using MHz XPCS. I believe the experiment described presents a compelling scientific case for XPCS at FELs, particularly with the use of EuXFEL pulse trains. However, there are several key aspects that need to be addressed before the paper can be considered for publication.

1. XFELs are known for the excellent transverse coherence, and this measurement was conducted at a very small angle. However, the observed contrast is only a few percent. What could be the reason for this? Please perform single-shot speckle visibility analysis to verify the contrast at each Q. Additionally, could you provide details on the theoretical calculations for the expected contrast values based on the measurement geometry?
2. In reference to question 1, Fig. 4 shows different starting contrast values for various concentrations at the same Q, while the starting contrast is a very important parameter in the fitting. It is hard to argue this discrepancy is due to the background. For example, at the smallest Q, the 70 mg/mL sample exhibits the highest scattering intensity, as shown in Fig. 1a, yet its contrast is not the highest. Why is that? Furthermore, since the contrast begins to decrease at the shortest separation (~220 ns), it becomes challenging to achieve a convincing fit without having the first data point at time zero from visibility extraction. For instance, the DLS measurement in Fig. S6a includes numerous short time g2 data points, which is what makes the fit reliable and accurate.
3. For figure 2, there should be error bars for each g2 values.
4. Regarding the error bars, the data points in Fig. 4b show significantly larger deviations from the theoretical model than suggested by the error bars, making it difficult to argue that the model is still valid.
5. Line 151 and Line 157, is the pre-factor (1.59) the same for hemoglobin and ferritin?
6. The authors should double check the numbers for consistency throughout the paper. From Fig. S6b, D_0 is about $3 \text{ nm}^2 \mu\text{s}^{-1}$, which differs from the value stated on line 277 (page 17) of the main paper. Moreover, using this D_0 , and the viscosity $\eta = 92 \text{ mPa s}$, I cannot get the correct R_h . D_0 of ferritin in water and the hydrodynamic radius makes sense on the other hand.
7. How is D_s/D_0 in Fig. 5b estimated? According to line 136 of the paper, $H(Q \gg q_0) = D_s/D_0$. However, when looking at Fig. 5a, the fit extending to $q = 0.8$ shows an increasing trend, suggesting that $H(q=0.8)$ does not represent the small length scale limit. Yet, the D_s/D_0 value in Fig. 5b appears to match $H(q = 0.8)$. It should also come with an error bar.
8. In this study, only the 730 mg/ml is studied in water, we see a considerate slow down considering the difference in τ_i (0.3 microsecond from stokes Einstein equation vs the measured interaction time of 4.25 microsecond). However, if using 55% vol fraction glycerol, is there a slow down? (D_1 seems to be within a factor of 2 as compared with D_0) What causes the difference in the magnitude of the change in interaction rate between these two environments? The data for 400mg/mL in water is needed as a comparison.

Reviewer #2

(Remarks to the Author)

The manuscript describes a comprehensive study on the diffusion of biomolecules in 'crowded' environment using megahertz X-ray photon correlation spectroscopy. Using ferritin as the model system, the authors revealed the cage-trapping phenomenon. This particular observation is relevant to understanding the diffusivity of ferritin, particularly - as mentioned in the manuscript - for its use as delivery system in biomedical applications.

The work was carried out comprehensively and shed lights on the understanding of how protein particles diffuse in solution. It will be interesting for the authors to comment on the cage-trapping behavior on other particles. Have they been observed on inorganic nanoparticles of similar size? What would be the parameters or surface properties that may drive such trapping?

While the reviewer appreciates the implications on biomedicine, ferritin has a unique property that each particle contains iron core that has been shown to be paramagnetic. This magnetic behavior is attractive for its application in MRI which the authors may want to include. How does the iron core affect the measurement? How does it play a role in the diffusivity of the ferritin?

Reviewer #3

(Remarks to the Author)

In a paper titled "Coherent X-rays reveal anomalous molecular diffusion and cage effects in crowded protein solutions", Girelli et al. investigate translational diffusion of ferritin in self-crowded systems using X-ray scattering. In particular, authors study the interplay of direct and hydrodynamic interactions, and timescales involved in the caging effects. The key result is a similar wavevector dependence of long- and short-time ferritin diffusivity, resulting in transferability of hydrodynamic function between these two regimes. The topic is very interesting and important for the biophysics and soft condensed matter community, and the findings are of high importance. Moreover, the methodology applied by the authors probes the microsecond timescale which was previously relatively unexplored.

The strength of this paper lies in the data which beyond any doubts is comprehensive (multiple techniques, multiple concentrations, multiple wavevectors) and accurate (small error bars, good agreement between model fits and experimental points). However, at the same time, multiple techniques and analysis approaches make it hard to follow and relate between the results of various subsections. Specifically, connection between various diffusion coefficients (D_{short} , D_{long} , D_1 , D_2) introduced in the paper and the temporal dynamics of mean squared displacement or concentration is not clearly expressed, which makes it hard to relate the findings with the problem of diffusive transport mentioned in the abstract. I would suggest improving these aspects. Below I list specific things that should be addressed.

1. Page 3, line 40: short-time diffusion should not be affected by the excluded volume, but by hydrodynamic interactions only.
2. Figure 4 is titled "Decorrelation rate (...)", but the symbol corresponds to average decorrelation rate introduced earlier. I believe this distinction is important and consistency should be kept.
3. In the caption of Figure 5b, after word "self-" the term "diffusion coefficient" is misplaced, or something is lacking.
4. The estimate of short-time diffusivity from two-component exponential fit to the intermediate scattering function is almost 15 times smaller than the dilute-regime diffusion coefficient, leading to drastically different estimates of transition time between the two different diffusive regimes. Authors show that theory predicts much smaller short-time diffusivity slowdown (dotted line in Fig. 5b). What is the reason for this discrepancy, and more general: what is the relation between the estimates made via this two-component exponential fit and the previous estimates made based on wavevector-dependent diffusivities?
5. Page 17, line 279: authors write viscosity = 92 mPa*S, which is 92 cP. It seems too high both for water and for water-glycerol mixture.
6. The major application of the results of this study mentioned by authors in the abstract is predicting diffusive transport of ferritin-based drugs. It is not clear to me how the various diffusion coefficients obtained in the paper translate into temporal dynamics, either mean squared displacement curve with its distinct regimes or temporal evolution of concentration field. Authors should shed some light on that.
7. Page 18, line 298: "correspond to standard errors"

Version 1:

Reviewer comments:

Reviewer #1

(Remarks to the Author)

I would like to thank the authors for their efforts in addressing my comments and suggestions. I still have a following up question on point 1 and 2. The details are attached.

Reviewer #2

(Remarks to the Author)

The authors have address all comments satisfactorily.

Version 2:

Reviewer comments:

Reviewer #1

(Remarks to the Author)

Rebuttal letter for "Coherent X-rays reveal anomalous molecular diffusion and cage effects in crowded protein solutions" (NCOMMS-24-55663)

We appreciate the time and effort of each of the three reviewers dedicated to providing insightful feedback on ways to strengthen our paper and the overall positive evaluation of our work. Here, we provide point-by-point responses to the reviewers' comments. The text is color-coded: The comments of the reviewers are colored black, our responses are colored green, and the modifications incorporated into the manuscript or the supporting information are colored blue.

Reviewer 1

This manuscript explores protein diffusion dynamics in a crowded environment using MHz XPCS. I believe the experiment described presents a compelling scientific case for XPCS at FELs, particularly with the use of EuXFEL pulse trains. However, there are several key aspects that need to be addressed before the paper can be considered for publication.

We thank the reviewer for providing interesting and detailed remarks about our work. We have revised our manuscript to address the comments of the reviewer, which helped significantly improve our article.

Point 1 XFELs are known for the excellent transverse coherence, and this measurement was conducted at a very small angle. However, the observed contrast is only a few percent. What could be the reason for this? Please perform single-shot speckle visibility analysis to verify the contrast at each q . Additionally, could you provide details on the theoretical calculations for the expected contrast values based on the measurement geometry?

We have added a single-shot speckle visibility analysis suggested by the referee. We note here that the values of the contrast are consistent with previous studies performed at the EuXFEL with similar experimental parameters [1–3]. As the referee points out, the transverse coherence is very high but the longitudinal is limited by the wavelength bandwidth, in accordance with contrast estimations [4]. In addition, in order to reduce the x-ray dose on the sample, the beam size is relatively large, which causes a further decrease of the speckle contrast. Furthermore, the contrast is limited by the relatively large size of the pixel of AGIPD ($200 \times 200 \mu m^2$), which is necessary to reach MHz resolution. We have added the following section "Contrast estimation" in the supporting information with details of the contrast estimation:

Contrast evaluation

The values of the contrast $\beta(q)$ were estimated by using the equation [1, 2]:

$$\beta(q) = \beta_l(q)\beta_t \quad (1)$$

The value of the transverse coherence was found to be $\beta_t \approx 0.5$ in various XFELs including the EuXFEL [1, 2, 5]. The longitudinal coherence, $\beta_l(q)$, describes the loss of coherence due to the finite bandwidth $\Delta E/E$, the geometry and speckle shape factor. $\beta_l(q) = (M_{\text{rad}}M_{\text{det}})^{-1}$ contains two contributions M_{rad} and M_{det} , which are:

$$M_{\text{rad}} = \sqrt{1 + \frac{q^2 \left(\frac{\Delta E}{E}\right)^2 [b_s^2 \cos^2(\theta) + t^2 \sin^2(\theta)]}{4\pi^2}} \quad (2)$$

$$M_{\text{det}} = \sqrt{1 + \frac{p^4 b_s^2 [b_s^2 \cos^2(2\theta) + t^2 \sin^2(2\theta)]}{\lambda^4 L^4 M_{\text{rad}}^2}}. \quad (3)$$

With the constant b_s being the beam size, $\Delta E/E$ being the bandwidth, 2θ being the scattering angle, λ being the wavelength of the x-ray beam, L being the sample-detector distance, p being the pixel size, t being the sample thickness. The experimental parameters in the fit were fixed, except for the beam size b_s which effectively accounts for small changes in the experimental parameter between measurements. The value of the beam size was optimized for each sample in the fit g_2 function. The corresponding extracted values of the contrast with are reported in Fig.S9 and their respective b_s in Table 4.

Figure S9: a) Speckle contrast β versus momentum transfer q , obtained after the optimization of the beam size in the fit of $g_2(q, t)$ from Fig. 3 of the main manuscript. The shaded area represents the standard error due to the uncertainties of the fit. b) Contrast estimation with three different methods for the data of protein concentration 180 mg/ml. The three methods correspond to: the fit results from the $g_2(q, t)$ functions (green circles and shaded area to indicate the standard error), the estimator of eq. 7, the fit of the probability of two photon events given in Eq. 6 (orange squares).

c (mg/ml)	b_s (μm)
70	13.4 ± 1.3
180	12.4 ± 0.3
400	12.8 ± 0.3
730	13.2 ± 0.4

Table 4: Beam size obtained from the fit of $g_2(q, t)$ from Fig. 3 of the main manuscript.

To independently verify the speckle contrast values obtained by the fit, we perform a single shot X-ray speckle visibility analysis (XSVS). Here, the contrast β can be obtained by analysing the properties of scattering intensity distribution $P(k, \bar{k}, M)$ of a single frame [6]. The analytical form of $P(k, \bar{k}, M)$ is:

$$P(k, \bar{k}, M) = \frac{\Gamma(k+M)}{\Gamma(M)\Gamma(k+1)} \left(1 + \frac{M}{\bar{k}}\right)^{-k} \left(1 + \frac{\bar{k}}{M}\right)^{-M} \quad (4)$$

where k is the photon count of each pixel per shot, \bar{k} is the average number of photons per pixel per shot and M is the number of modes, which is related to the speckle contrast by $\beta = 1/M$. The contrast was estimated with two different methods: from the fit of the probability of finding a one- and two-photon events on a pixel $p_2 = P(k=2, \bar{k}, M)$ and from a contrast estimator formula. The

analytical form for p_1 and p_2 can be derived from Eq. 4, and is:

$$p_1 = P(k = 1, \bar{k}, M) = M \left(1 + \frac{M}{\bar{k}}\right)^{-1} \left(1 + \frac{\bar{k}}{M}\right)^{-M}, \quad (5)$$

$$p_2 = P(k = 2, \bar{k}, M) = \frac{M(M+1)}{2} \left(1 + \frac{M}{\bar{k}}\right)^{-2} \left(1 + \frac{\bar{k}}{M}\right)^{-M}. \quad (6)$$

From the analytical expression of p_1 and p_2 , and defining R_{12} as $R_{12} = p_2/p_1$, β can be obtained with the following estimator[7]:

$$\beta = \frac{2 \cdot R_{12} - \bar{k}}{\bar{k}(1 - 2 \cdot R_{12})} \quad (7)$$

The contrast values were obtained using the single images of pulses from 1 to 120 and the exact same bunch trains used for the calculation of $g_2(q, t)$. We observe that the contrast values obtained with different methods are overall consistent with each other. The minor discrepancies between the contrast obtained from the fit of the TTCs and from the XSVS analysis can be due to several reasons: First, in the XSVS analysis it is not straightforward to apply a cross-correlation correction [8] which corrects for the spurious correlations present due to the noise in the detector. Second, in the single image XSVS analysis a baseline correction is not trivial as it is for the two-time correlation functions, where the correlation of images at large time difference can be used as baseline. For XSVS, the large error bars are due to the very low counts which makes the baseline estimation challenging [9].

Point 2 In reference to question 1, Fig. 4 shows different starting contrast values for various concentrations at the same q , while the starting contrast is a very important parameter in the fitting. It is hard to argue this discrepancy is due to the background. For example, at the smallest q , the 70 mg/mL sample exhibits the highest scattering intensity, as shown in Fig. 1a, yet its contrast is not the highest. Why is that? Furthermore, since the contrast begins to decrease at the shortest separation (220 ns), it becomes challenging to achieve a convincing fit without having the first data point at time zero from visibility extraction. For instance, the DLS measurement in Fig. S6a includes numerous short time g_2 data points, which is what makes the fit reliable and accurate.

As the referee point out, the contrast does indeed change between different measurements and runs although this is mainly due to the changes of the XFEL beam properties, and not due to the scattering signal of the sample. To account for the variation in the contrast, and the limitation to 222.2 ns for the shortest time delay, a theoretical estimation of the contrast has been included in the fit of the $g_2(q, t)$. The model takes into account the q dependence of the contrast based on the experimental parameters (X-ray bandwidth, sample-detector distance, photon energy, sample thickness etc.) based on Refs.[1, 2]. In order to take into account changes in contrast, the beam size is fitted along with the decorrelation rate $\tilde{\Gamma}$. The additional beam parameters used for the estimation are reported in Table 3. We have added the following section in the Methods section, under "Intensity autocorrelation function g_2 calculation and fit":

The fits of the exponential functions to $g_2(q, t)$ were performed simultaneously in both q and t dimensions, to ensure better stability of the fits. The fit contains the q -dependence and the overall values of the contrast $\beta(q)$ as described in SI in section "Contrast estimation". The contrast $\beta(q)$ was included to overcome the limitation of 222.2 ns for the shortest time delay. The experimental parameters for the contrast estimation (X-ray bandwidth, sample-detector distance, photon energy, sample thickness etc.) were fixed with the values reported in Table 3 except for the beam size. The latter was set as fit parameter to account for the inherent fluctuations of the XFEL beam (for values, see the SI section "Contrast estimation"). The value of α was set to be constant with q . The q -dependence of $\tilde{\Gamma}(q)$ was not fixed to be able to extract $D(q)$.

Point 3 For figure 2, there should be error bars for each g_2 values.

The errorbar are now included for all g_2 functions and if not visible, they are smaller than the symbol size. Fig.3 of the main manuscript was changed.

Figure 3: **X-ray Photon Correlation Spectroscopy (XPCS) data of ferritin solutions obtained at EuXFEL.** The intensity autocorrelation function, $g_2(q, t)$, for different protein concentrations (a) $c = 70$ mg/ml, (b) $c = 180$ mg/ml, (c) $c = 400$ mg/ml and (d) $c = 730$ mg/ml. Data in panels (a-c) were measured in water-glycerol (with glycerol volume fraction $\nu_{\text{glyc}} = 0.55$), while panel (d) presents data measured in water to reach the desired protein concentration ($c = 730$ mg/ml). The different colors represent different q -values, changing from lighter to darker green for q increasing from $q = 0.225$ nm $^{-1}$ to $q = 0.625$ nm $^{-1}$ with equal spacing of $dq = 0.05$ nm $^{-1}$. Solid lines represent stretched exponential fits, with the corresponding Kohlrausch-Williams-Watts (KWW) exponent α shown in the legend. The error bars shown correspond to the standard error, estimated as described in the SI.

Point 4 Regarding the error bars, the data points in Fig. 4b show significantly larger deviations from the theoretical model than suggested by the error bars, making it difficult to argue that the model is still valid.

To improve the stability of the fit, we have introduced a baseline correction in the TTCs. It is true that, as the referee points out, we observe deviations from the theoretical model, but we note here that our data provide a significant improvement from previous measurements. [10–12]. In Methods, in section "Intensity autocorrelation function g_2 calculation and fit" the following sentence was added: A baseline subtraction on the average TTCs was performed at each q . The value of the baseline was calculated as weighted mean of the delay times correspondent to pulses 210 to 310, corresponding to delay times above 46 μ s.

Point 5 Line 151 and Line 157, is the pre-factor (1.59) the same for hemoglobin and ferritin?

We thank the reviewer for pointing out that the possible confusion in the phrasing. In case of hemoglobin the pre-factor is 1.36. We have changed it to:

In the case of hemoglobin [12], this discrepancy was partially explained by accounting for the hydrodynamic volume fraction.

Point 6 The authors should double check the numbers for consistency throughout the paper. From Fig. S6b, D_0 is about $3 \text{ nm}^2 \mu\text{s}^{-2}$, which differs from the value stated on line 277 (page 17) of the main paper. Moreover, using this D_0 , and the viscosity $\eta = 92 \text{ mPas}$, I cannot get the correct R_h . D_0 of ferritin in water and the hydrodynamic radius makes sense on the other hand.

We thank the reviewer for noticing the discrepancy in the values. They are now corrected. In Methods, the section "Brownian diffusion coefficient D_0 and hydrodynamic radius R_h " now reads:

The protein diffusion coefficient, D_0 , was measured in dilute conditions ($c = 9 \text{ mg/ml}$) using DLS (see Section 4 of the Supplementary Information). For ferritin in water-glycerol (55 vol% glycerol), we obtained $D_0 = 3.05 \pm 0.01 \text{ nm}^2 \mu\text{s}^{-1}$. For ferritin in water the diffusion coefficient used is $D_0 = 30.1 \pm 0.2 \text{ nm}^2 \mu\text{s}^{-1}$. The hydrodynamic radius, R_h , was calculated using the Stokes-Einstein equation, $D_0 = k_B T / (6\pi\eta R_h)$, where η is the viscosity, T the temperature and k_B the Boltzmann constant. Here, for water-glycerol by using $\eta = 9.3 \pm 0.6 \text{ mPas}$ [13] and $T = 298 \text{ K}$, we estimated $R_h = 7.3 \pm 0.1 \text{ nm}$.

Point 7 How is D_s/D_0 in Fig. 5b estimated? According to line 136 of the paper, $H(Q \gg q_0) = D_s/D_0$. However, when looking at Fig. 5a, the fit extending to $q = 0.8$ shows an increasing trend, suggesting that $H(q = 0.8)$ does not represent the small length-scale limit. Yet, the D_s/D_0 value in Fig. 5b appears to match $H(q = 0.8)$. It should also come with an error bar.

We have added Figure S7 in the SI in section " $H(q)$ model fit" which shows the extended q -range, as well as explanation in the main text to clarify this point:

The extracted D_s/D_0 , estimated as the average value $D(q)S(q)/D_0$ in a q range between 0.8 and 1.3 nm^{-1} , as a function of the hydrodynamic volume fraction, is shown in Fig. 5b (see also Fig. S7).

Figure S7: The experimental $D(q)S(q)/D_0$ as a function of momentum transfer q (empty symbols) and $H(q)$ model results using the $\delta\gamma$ -theory (solid lines). The data points and lines are the same as in Fig. 5a, but the q range was extended to show the oscillations of the hydrodynamic function around the estimated D_s/D_0 . The figure is shown in a) linear scale and in b) log scale to be able to see the data points for the highest protein concentration.

In the caption of Fig.5 we added: The error bars correspond to error propagation from the fit parameters and, if not visible, they are smaller than the symbol.

Point 8 In this study, only the 730 mg/ml is studied in water, we see a considerable slow down considering the difference in τ_i (0.3 microsecond from stokes Einstein equation vs the measured interaction

time of 4.25 microsecond). However, if using 55% vol fraction glycerol, is there a slow down? (D_1 seems to be within a factor of 2 as compared with D_0) What causes the difference in the magnitude of the change in interaction rate between these two environments? The data for 400 mg/mL in water is needed as a comparison.

The referee raises an important point. A direct comparison of ferritin in water at low and intermediate concentration is not possible with MHz-XPCS in the current setup, where the shortest time delay is 222.2 ns. This can be extrapolated from the current data we collected on ferritin at 400 mg/ml in water-glycerol (Fig. 3). Taking into account the difference in viscosity (factor of ~ 10), most of the decay of $g_2(q, t)$ for ferritin in water is outside the experimental window. However, the diffusion of ferritin in heavy water at intermediate and low concentration was previously extracted with simulations and neutron scattering experiments [10]. We have added the following section to clarify this issue under "Cage effects and long-time diffusion":

The short-time diffusion coefficient $D^{short}(q)$ is almost 15 times smaller than the dilute-regime diffusion coefficient. The reason for the discrepancy between $D^{short}(q)$ and the dotted line in Fig. 5b is due to the limit of the validity of Eq. 4 to volume fractions below 0.45 according to Ref.[14]. For ferritin in heavy water at $\phi = 0.153$ and $\phi = 0.327$ (corresponding to ϕ_h of 0.21 and 0.44), the values of the normalized short-time self-diffusion D_s/D_0 , extracted with simulations and experimental data employing neutron scattering [10], are 0.76 and 0.48 respectively. These values are close to the expected short-time diffusion coefficient (dotted line in Fig.5b). Hence, we conclude that the large difference seen between the samples in glycerol-water mixtures and water cannot be attributed to the different solvents, but rather to very elevated crowding conditions of the proteins in solution at 730 mg/ml.

Reviewer 2

The manuscript describes a comprehensive study on the diffusion of biomolecules in 'crowded' environment using megahertz X-ray photon correlation spectroscopy. Using ferritin as the model system, the authors revealed the cage-trapping phenomenon. This particular observation is relevant to understanding the diffusivity of ferritin, particularly - as mentioned in the manuscript - for its use as delivery system in biomedical applications. The work was carried out comprehensively and shed lights on the understanding of how protein particles diffuse in solution.

We thank the reviewer for providing remarks about the possible applications of our work. We have revised our manuscript to address the comments of the reviewer, which helped to deepen the clarity of our article and connect it more with applications and previous studies.

Point 1 It will be interesting for the authors to comment on the cage-trapping behavior on other particles. Have they been observed on inorganic nanoparticles of similar size? What would be the parameters or surface properties that may drive such trapping?

We thank the referee for raising these questions. In section "Cage effects and long-time diffusion" we have added the following sentences:

This result aligns with the expected reduction in cage size due to confinement with increasing volume fraction, approaching the glass transition [15–17]. Cage effects have been probed also in biological systems [18, 19] and inorganic colloids with different sizes [17, 20]. Here, the main parameter that drives the cage formation is the excluded volume effects, which can be affected by the inter-particle interactions [21].

Point 2 While the reviewer appreciates the implications on biomedicine, ferritin has a unique property that each particle contains iron core that has been shown to be paramagnetic. This magnetic behavior is attractive for its application in MRI which the authors may want to include. How does the iron core affect the measurement? How does it play a role in the diffusivity of the ferritin?

We thank the reviewer for pointing out the important application in MRI, we have included this possible application in the text. In our experiment the diffusion coefficient is specifically related to the probed length scale associated with the average protein-protein distance, which is larger than the core itself. However, there can be smaller indirect contributions given that the hydrodynamic radius of ferritin and apoferritin is different, as indicated previously [11]. We have added the following section in the main manuscript:

Furthermore, ferritin has been suggested as an MRI contrast agent given the paramagnetic properties of its core [22], and the possibility of exchanging the content of the core [23]. In this context, ferritin aggregation may be a useful strategy to produce a functional reporter gene for magnetic resonance imaging [24].

Reviewer 3

In a paper titled “Coherent X-rays reveal anomalous molecular diffusion and cage effects in crowded protein solutions”, Girelli et al. investigate translational diffusion of ferritin in self-crowded systems using X-ray scattering. In particular, authors study the interplay of direct and hydrodynamic interactions, and timescales involved in the caging effects. The key result is a similar wavevector dependence of long- and short-time ferritin diffusivity, resulting in transferability of hydrodynamic function between these two regimes. The topic is very interesting and important for the biophysics and soft condensed matter community, and the findings are of high importance. Moreover, the methodology applied by the authors probes the microsecond timescale which was previously relatively unexplored.

The strength of this paper lies in the data which beyond any doubts is comprehensive (multiple techniques, multiple concentrations, multiple wavevectors) and accurate (small error bars, good agreement between model fits and experimental points). However, at the same time, multiple techniques and analysis approaches make it hard to follow and relate between the results of various subsections. Specifically, connection between various diffusion coefficients (D_{short} , D_{long} , D_1 , D_2) introduced in the paper and the temporal dynamics of mean squared displacement or concentration is not clearly expressed, which makes it hard to relate the findings with the problem of diffusive transport mentioned in the abstract. I would suggest improving these aspects. Below I list specific things that should be addressed.

We thank the reviewer for the attentive reading and detailed remarks about our work. We have revised our manuscript to improve the readability and added more explanations about the interpretation and understanding of our data. We have changed the names from D_1 and D_2 to D^{short} and D^{long} to improve the clarity of the text. We have also provided more details on the connection of the measured quantities with mean squared displacement in the SI as well as in the text (see point 7).

Point 1 Page 3, line 40: short-time diffusion should not be affected by the excluded volume, but by hydrodynamic interactions only.

We thank the reviewer for pointing this out, we have changed the wording. The sentence was changed to “by hydrodynamic interactions.”

Point 2 Figure 4 is titled “Decorrelation rate (...)”, but the symbol corresponds to average decorrelation rate introduced earlier. I believe this distinction is important and consistency should be kept. We thank the reviewer for pointing out the inconsistency, it is now correct.

Point 3 In the caption of Figure 5b, after word “self-” the term “diffusion coefficient” is misplaced, or something is lacking.

We thank the reviewer for pointing out the inconsistency, we have corrected it.

Point 4 The estimate of short-time diffusivity from two-component exponential fit to the intermediate scattering function is almost 15 times smaller than the dilute-regime diffusion coefficient, leading to drastically different estimates of transition time between the two different diffusive regimes. Authors show that theory predicts much smaller short-time diffusivity slowdown (dotted line in Fig. 5b). What is the reason for this discrepancy, and more general: what is the relation between the estimates made via this two-component exponential fit and the previous estimates made based on wavevector-dependent diffusivities?

We thank the referee for pointing out the lack of clarity of these two points. We have answered the questions and addressed the points by adding the two following paragraphs in section “Cage effects and long-time diffusion” :

The short-time diffusion coefficient $D^{short}(q)$ is almost 15 times smaller than the dilute-regime diffusion coefficient. The reason for the discrepancy between $D^{short}(q)$ and the dotted line in Fig. 5b is due to the limit of the validity of Eq. 4 to volume fractions below 0.45 according to Ref. [14].

Here, we have used two models to extract the diffusion coefficient: the stretched exponential and the double exponential models. In the stretched exponential the short- and long-time diffusion are effectively merged into a single time constant, and the exponent indirectly reflects the extent to which these two components are separated as a function of concentration. In the double exponential model, the long- and short-time diffusion are treated separately and the amplitude indicates the relative contributions of the two components. The latter indicates that 90% of the decay is from the long-time diffusion component. Since the contribution of long-time diffusion component is predominant, the extracted value of $D(q)$ obtained from the stretched exponential model is effectively close to the value of $D^{long}(q)$. The solid black line shown in Fig. 4b and in Fig. 6c (it refers to the y-axis on right-hand side) is the same in the two panels, showing good agreement between $D(q)$ and $D^{long}(q)$.

Point 5 Page 17, line 279: authors write viscosity = 92 mPa s, which is 92 cP. It seems too high both for water and for water-glycerol mixture.

We thank the reviewer for pointing out this discrepancy, we have corrected this (see also response to reviewer 1 Point 6).

Point 6 The major application of the results of this study mentioned by authors in the abstract is predicting diffusive transport of ferritin-based drugs. It is not clear to me how the various diffusion coefficients obtained in the paper translate into temporal dynamics, either mean squared displacement curve with its distinct regimes or temporal evolution of concentration field. Authors should shed some light on that.

We have added the section "Mean square displacement and intermediate scattering function" in the supporting information:

The intensity autocorrelation function and the intermediate scattering function are related to both the mean square displacement of the particle and the time dependent concentration fluctuations. The width function, defined as

$$W(q, t) = -\frac{\ln[f(q, t)]}{q^2}, \quad (8)$$

corresponds to the mean square displacement $\langle r^2(t) \rangle = W(q, t)$ in the case of a dilute system. In case of interacting particles this relation does not hold, as the mean-square displacement refers only to the self-part of the intermediate scattering function. In other words, $\langle r^2(t) \rangle$ includes only self-diffusion and not the contribution of collective diffusion. In case of interacting particles, only in the limit of $q \gg q_0$ the intermediate scattering function does not contain contributions from the collective diffusion, and therefore Eq. 8 holds.

For low protein concentration the relation between the self-diffusion coefficient and the mean square displacement is

$$\langle r^2(t) \rangle = 6D_s t. \quad (9)$$

The presence of two exponential decays indicates that in the mean square displacement is not showing a simple relation as seen in Eq. 9. Instead, two different slopes are present, one D_s^{short} corresponding to time $t \ll \tau_i$ and the other one D_s^{long} at $t \gg \tau_i$.

In the introduction we have added the following sentence:

These two diffusion types are visible also by probing the mean square displacement (MSD) of the particles. They are characterized by a linear increase of MSD with time with one slope at $t \ll \tau_i$ and a different one $t \gg \tau_i$ [25]

In the section "Cage effects and long-time diffusion" the following sentence was added:

For Brownian diffusion in the dilute limit, $f(q, t)$ is expected to be a simple exponential function, meaning that $\ln[f(q, t)]$ depends linearly on time, and $\ln[f(q, t)] = -\langle r^2(t) \rangle q^2$ with $\langle r^2(t) \rangle$ the mean square displacement.

Point 7 Page 18, line 298: "correspond to standard errors"
We thank the reviewer for pointing out typo. It is corrected now.

References

1. Lehmkuhler, F. *et al.* Emergence of Anomalous Dynamics in Soft Matter Probed at the European XFEL. *Proc. Natl. Acad. Sci. U.S.A.* **117**, 24110–24116 (2020).
2. Reiser, M. *et al.* Resolving molecular diffusion and aggregation of antibody proteins with megahertz X-ray free-electron laser pulses. *Nat. Commun.* **13**, 5528 (2022).
3. Dallari, F. *et al.* Coherence properties from speckle contrast analysis at the European XFEL. *J. Phys. Conf. Ser.* **2380**, 012085 (2022).
4. Perakis, F. & Gutt, C. Towards molecular movies with X-ray photon correlation spectroscopy. *Phys. Chem. Chem. Phys.* **22**, 19443–19453 (2020).
5. Madsen, A. *et al.* Materials Imaging and Dynamics (MID) instrument at the European X-ray Free-Electron Laser Facility. *J. Synchrotron Radiat.* **28**, 637–649 (2021).
6. Hruszkewycz, S. O. *et al.* High Contrast X-ray Speckle from Atomic-Scale Order in Liquids and Glasses. *Phys. Rev. Lett.* **109**, 185502 (2012).
7. Perakis, F. *et al.* Coherent X-rays reveal the influence of cage effects on ultrafast water dynamics. *en. Nat. Commun.* **9**, 1917 (2018).
8. Dallari, F. *et al.* Analysis Strategies for MHz XPCS at the European XFEL. *Appl. Sci.* **11**, 8037 (2021).
9. Möller, J. *et al.* Using low dose X-ray Speckle Visibility Spectroscopy to study dynamics of soft matter samples. *New J. Phys.* **23**, 093041 (2021).
10. Gapinski, J. *et al.* Diffusion and microstructural properties of solutions charged nanosized proteins: experiment versus theory. *J. Chem. Phys.* **123**, 054708 (2005).
11. Häussler, W. Neutron spin echo studies on ferritin: free-particle diffusion and interacting solutions. *Eur. Biophys. J.* **37**, 563–571 (2008).
12. Doster, W. & Longeville, S. Microscopic Diffusion and Hydrodynamic Interactions of Hemoglobin in Red Blood Cells. *Biophys. J.* **93**, 1360–1368 (2007).
13. Cheng, N.-S. Formula for the Viscosity of a GlycerolWater Mixture. *Ind. Eng. Chem. Res.* **47**, 3285–3288 (2008).
14. Beenakker, C. Self-diffusion of spheres in a concentrated suspension. *Physica A* **120**, 388–410 (1983).
15. Weeks, E. R. & Weitz, D. A. Properties of Cage Rearrangements Observed near the Colloidal Glass Transition. *Phys. Rev. Lett.* **89**, 095704 (2002).
16. Weeks, E. R., Crocker, J. C., Levitt, A. C., Schofield, A. & Weitz, D. A. Three-Dimensional Direct Imaging of Structural Relaxation Near the Colloidal Glass Transition. *Science* **287**, 627–631 (2000).
17. Segrè, P. N. & Pusey, P. N. Scaling of the Dynamic Scattering Function of Concentrated Colloidal Suspensions. *Phys. Rev. Lett.* **77**, 771–774 (1996).
18. Chushkin, Y. *et al.* Probing cage relaxation in concentrated protein solutions by XPCS. *Phys. Rev. Lett.* **129**, 238001 (2022).
19. Goiko, M., de Bruyn, J. R. & Heit, B. Short-Lived Cages Restrict Protein Diffusion in the Plasma Membrane. *Sci. Rep.* **6**, 34987 (2016).
20. Kwaśniewski Paweł and Fluerasu, A. & Madsen, A. Anomalous dynamics at the hard-sphere glass transition. *Soft Matter* **10**, 8698–8704 (43 2014).
21. Kompella, V. P. S., Romano, M. C., Stansfield, I. & Mancera, R. L. What determines sub-diffusive behavior in crowded protein solutions? *Biophys. J* **123**, 134–146 (2024).

22. Gossuin, Y., Roch, A., Muller, R. N., Gillis, P. & Lo Bue, F. Anomalous nuclear magnetic relaxation of aqueous solutions of ferritin: An unprecedented first-order mechanism. *Magn. Reson. Med.* **48**, 959–964 (2002).
23. Ruggiero, M. R., Alberti, D., Bitonto, V. & Geninatti Crich, S. Ferritin: A Platform for MRI Contrast Agents Delivery. *Inorganics* **7** (2019).
24. Bennett, K. M., Shapiro, E. M., Sotak, C. H. & Koretsky, A. P. Controlled Aggregation of Ferritin to Modulate MRI Relaxivity. *Biophys. J.* **95**, 342–351 (2008).
25. Höfling, F. & Franosch, T. Anomalous transport in the crowded world of biological cells. *Rep. Prog. Phys* **76**, 046602 (2013).

Follow-up question on Manuscript "Coherent X-rays reveal anomalous molecular diffusion and cage effects in crowded protein solutions"

I would like to thank the authors for their efforts in addressing my comments and suggestions.

I still have a following up question on point 1 and 2. Understood that the g2 analysis utilizes the cross correlation as described in Ref 8, the excessive correlation due to detector artifacts as extracted from the cross-correlation analysis in principle can be directly used for correcting the visibility extraction. Below I briefly wrote my reasoning.

Say the intensity from pulse 1 and pulse 2 is I_1 and I_2 (pulse 1 and pulse 2 can be from different trains and sample is decorrelated), the detector artifact is I_a , the cross correlation gives

$$C = \frac{\langle (I_1 + I_a)(I_2 + I_a) \rangle}{\langle I_1 + I_a \rangle \langle I_2 + I_a \rangle} = \frac{\langle I_a^2 \rangle}{\langle I_1 + I_a \rangle^2}. \quad (1)$$

My understanding is that this is how the cross correlation method works. Then in principle, for single beam contrast

$$\beta_{0,extracted} = \frac{\langle (I_1 + I_a)(I_1 + I_a) \rangle}{\langle I_1 + I_a \rangle^2} - 1 = \frac{\langle I_1^2 \rangle}{\langle I_1 + I_a \rangle^2} + \frac{\langle I_a^2 \rangle}{\langle I_1 + I_a \rangle^2} - 1 \quad (2)$$

I guess it might be just a simple subtraction, which also agrees with the fact that the current contrast estimator is overestimating.

Answer to reviewer 1

We thank the reviewer for their comment. Their explanation would indeed be accurate if the photon counts were sufficiently high. However, in our case, the probability of finding a pixel with an intensity greater than 0 in a single pulse is approximately 0.01 (depending on the sample and the q range see Fig.1). Therefore, it is not possible to calculate the contrast using the formula suggested.

Fig. 1: The probability to find one and two photon events per pixel at different q values for a Ferritin concentration of 180 mg/ml for the q values indicated above the panels.

Furthermore, I_a is not a Gaussian or a Poissonian random variable, and it depends specifically which depends on storage cell in the detector and time (we cannot increase the statistics by probing for longer time to apply the central limit theorem and treat it as a gaussian variable). Hence, including its presence in Eq. S5 and consequently S7 and S8 is not trivial. This is visible also from the fit of p_2 which is not perfectly describing the data.

Answer to reviewer 1

We thank the reviewer for their comment. Their explanation would indeed be accurate if the photon counts were sufficiently high. However, in our case, the probability of finding a pixel with an intensity greater than 0 in a single pulse is approximately 0.01 (depending on the sample and the q range see Fig.1). Therefore, it is not possible to calculate the contrast using the formula suggested.

Fig. 1: The probability to find one and two photon events per pixel at different q values for a Ferritin concentration of 180 mg/ml for the q values indicated above the panels.

Furthermore, I_a is not a Gaussian or a Poissonian random variable, and it depends specifically which depends on storage cell in the detector and time (we cannot increase the statistics by probing for longer time to apply the central limit theorem and treat it as a gaussian variable). Hence, including its presence in Eq. S5 and consequently S7 and S8 is not trivial. This is visible also from the fit of p_2 which is not perfectly describing the data.

I thank the reviewer for answer. However, there might be a misunderstanding in what I suggested.

First, in this shot noise dominated regime, the statistics are well described by the negative binomial distribution, from which one can get the extracted contrast $\beta_{0,extracted}$ from a certain number of pixels and detector frames.

Second, the equation (2) in my previous comment only relies on the assumption that the detector artifact is uncorrelated with the shot noise, thus equation (2) will work if we can characterize $\langle I_a^2 \rangle$.

Since g_2 analysis can be corrected using cross-correlation between pulse trains, this implies that the artifact appears consistently at certain pulse numbers within each train, likely due to the AGIPD readout scheme. I believe such a behavior, if it drifts, also takes place on second or longer timescales to allow this correction.

This suggests that the same cross-correlation value ($1 + \Delta$) used in g_2 correction can be applied to correct the additional term in the speckle visibility. Therefore, subtracting Δ from $\beta_{0,extracted}$ may provide a more accurate estimate of the true contrast. One important thing is that the pulse IDs (within a train) used to extract Δ from cross-correlation must match those used in the contrast analysis.

On the other hand, I believe this excess term does not affect the outcome of the g_2 fitting, which was the original motivation behind my comment. Given that including or excluding the “time zero” point does not lead to a significant change in the fit, I find the scientific conclusions of the manuscript to be well supported and coherent. I therefore agree that the manuscript is suitable for acceptance.

Referee 1

I thank the reviewer for answer. However, there might be a misunderstanding in what I suggested.

First, in this shot noise dominated regime, the statistics are well described by the negative binomial distribution, from which one can get the extracted contrast $\beta_{0,\text{extracted}}$ from a certain number of pixels and detector frames.

Second, the equation (2) in my previous comment only relies on the assumption that the detector artifact is uncorrelated with the shot noise, thus equation (2) will work if we can characterize. Since g_2 analysis can be corrected using cross-correlation between pulse trains, this implies that the artifact appears consistently at certain pulse numbers within each train, likely due to the AGIPD readout scheme. I believe such a behavior, if it drifts, also takes place on second or longer timescales to allow this correction.

This suggests that the same cross-correlation value $(1 + \Delta)$ used in g_2 correction can be applied to correct the additional term in the speckle visibility. Therefore, subtracting Δ from it may provide a more accurate estimate of the true contrast. One important thing is that the pulse IDs (within a train) used to extract Δ from cross-correlation must match those used in the contrast analysis.

On the other hand, I believe this excess term does not affect the outcome of the g_2 fitting, which was the original motivation behind my comment. Given that including or excluding the “time zero” point does not lead to a significant change in the fit, I find the scientific conclusions of the manuscript to be well supported and coherent. I therefore agree that the manuscript is suitable for acceptance.

Comments to referee 1

We thank the referee for their comments. We agree with them, including or excluding the “timezero” point does not lead to a significant change in the fit results with the fitting routine we used since the contrast is fitted with the majority of the experimental parameters fixed, as described in the SI.